# Simulating Hypoxia in a New England Estuary: WASP8 Advanced Eutrophication Module (Narragansett Bay, RI, USA)

Christopher D. Knightes 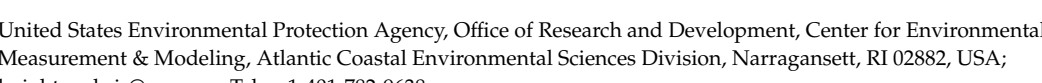

United States Environmental Protection Agency, Office of Research and Development, Center for Environmental Measurement & Modeling, Atlantic Coastal Environmental Sciences Division, Narragansett, RI 02882, USA; knightes.chris@epa.gov; Tel.: +1-401-782-9638

**Abstract:** Anthropogenic sources of nutrients cause eutrophication in coastal waters. Narraganset Bay (USA), the largest estuary in New England, has large seasonal zones of hypoxia. In response, management strategies have been implemented to reduce nutrient loadings. In this study, a mechanistic, mass balance fate and transport modeling framework was developed and applied to Narragansett Bay to improve our understanding of the processes governing hypoxia. Discrete and continuous observations were used for model comparison and evaluation. Simulations captured the general trends and patterns in dissolved oxygen (DO) with depth and space. Simulations were unable to capture the wide diurnal range of observed continuous DO and phytoplankton concentrations, potentially suggesting the need for improved understanding of processes at this time scale. Mechanistic modelling scenarios were performed to investigate how different sources of nutrients affect DO. Results suggest tributary sources of nitrogen affected upper layers of DO, while sediment oxygen demand and nutrient fluxes affected deeper waters. This work highlights the importance of understanding and simulating the legacy effects of historic nutrient loading to estuary systems to understand the magnitude and timing of long-term recovery due to reductions in nutrient loadings.

**Keywords:** surface waters; numerical modeling; nutrients; Narragansett Bay; nitrogen; dissolved oxygen; hypoxia; WASP





## 1. Introduction

Estuaries are critically important aquatic ecosystems. Estuaries provide a range of ecosystem services including biodiversity, transportation, flood and storm protection, primary production, recreation, and food [1–3]. Because they receive both freshwater and marine inputs, estuaries provide a transition zone which creates an environment particularly favorable for a variety of wildlife and fisheries, contributing to the economy of coastal areas [4]. Water quality is impacted by the land use and land cover of the surrounding watershed. Anthropogenic LULC change, including urbanization and agricultural development, impacts hydrology, erosion, and nutrient and contaminant loading to receiving waters [5–7]. Estuaries are primary sites for human activity as they are strategic locations for settlement and development, providing access to fresh water, surrounding lands for agricultural development, as well as river access to uplands, and productivity from fish and shellfish [2]. Urbanization has compromised estuaries as growing populations put increasing pressures on the ecosystem, causing deterioration in water quality and degradation to aquatic habitats. Excess nutrients coming from anthropogenic non-point and point sources have resulted in eutrophication of estuaries, resulting in zones of hypoxia (low dissolved oxygen (DO)) [8,9].

Dissolved oxygen is one of the most important ecological parameters in coastal aquatic ecosystems [10]. The number and extent of anoxic (DO = 0 mg L$^{-1}$) and hypoxic (DO < 2 mg L$^{-1}$) zones have demonstrably increased over the decades [11]. The increased load of anthropogenic sources of nutrients to surface waters has additionally

resulted in increased harmful algal blooms (HABs), decreased water clarity, and loss of aquatic habitats [11]. Hypoxia has been demonstrated in coastal systems all around the world [12–16].

Narraganset Bay ("the Bay") is the largest estuary in New England, and supports economic, recreation, and tourism [17]. Large zones of hypoxia have been observed in the northern portion of the Bay and into the tributaries feeding the Bay [17,18]. DO in the northern part of Narragansett Bay and Providence River have been estimated at less than 2.3 mg L$^{-1}$, demonstrating acute hypoxia, in the summer of 2001 [16]. Eutrophication and hypoxia of the Bay have had ecological impacts as demonstrated by decreased biodiversity, abundance, and biomass of benthic communities [18]. Over the past decade, the State of Rhode Island has implemented reductions in nitrogen releases from wastewater treatment plants (WWTPs), which has resulted in nitrogen loads into the Bay decreasing by over 50%, with associated reductions in eutrophication [19].

Mechanistic models are powerful tools to investigate eutrophication in coastal systems. These models allow us to provide a systems-level perspective, by incorporating different processes governing eutrophication dynamics within an estuary. The combination of observations and mechanistic modeling allows us to synthesize information and provide simulated concentrations with time and space through the model domain for the period of interest [20]. Mechanistic models have been applied at a range of locations to address different questions. In the Chesapeake Bay, a 3-D hydrodynamic-biogeochemical model, based on the Regional Ocean Modeling System linked with the Estuarine Carbon Biogeochemistry Model has been developed to provide current conditions of dissolved oxygen on the km scale [21]. In the Northern Gulf of Mexico, a 3-D model linking the Navy Coastal Ocean Model to the Coastal Generalized Ecosystem Model was used to interpret and investigate hypoxic waters nearshore [22]. In the Neuse River Estuary, North Carolina, United States of America (US), the Environmental Fluid Dynamics Code (EFDC) was linked to the Water Quality Analysis Simulation Program (WASP6) to develop a 3-D mechanistic model to support the development of a Total Maximum Daily Load (TMDL) management strategy for eutrophication [23]. More recently, a hybrid mechanistic and Bayesian inference approach was used in the Neuse River to characterize process and predictive uncertainty for possible management strategies to reduce hypoxia in the system [24]. The application and success of these efforts show the utility of mechanistic modeling to improve our understanding of these systems and the feasibility of developing management strategies to address hypoxia. Given Rhode Island has implemented reductions of nitrogen from WWTPs into the Bay, there is interest in understanding how these reductions translate to improving water quality in the Bay [25]. One way to investigate this is via mechanistic modeling.

In this study, a mechanistic, differential mass-balance, fate and transport model was developed and used to simulate eutrophication dynamics in the Bay for the year 2009. The year 2009 was chosen because it was a wet year and one of the worst years for hypoxia in the Bay [17]. A 3-D hydrodynamic model was used to provide spatial and temporal flow information to account for circulation and stratification. Output from the hydrodynamic model is used to drive a water quality model. A 3-D water quality was developed and applied to simulate nutrients, DO, and phytoplankton for the simulated year. Simulated DO and chl *a* were compared to discrete and continuous observations. Qualitative and quantitative metrics were used to investigate model performance. The Bay WASP model is then used to evaluate the feasible impacts of different sources of nutrients within the Bay on DO by performing mechanistic modeling scenarios.

The goal of this work is to simulate the changing DO and phytoplankton concentrations (as chlorophyll *a*, chl *a*) through the spatial domain of the Bay and with depth for the given year to improve our understanding of the processes governing zones of hypoxia and phytoplankton growth. The Bay has several islands, resulting in different flow paths and circulation patterns, unlike some other studied estuaries, which are dominated by open waters. In this study, the hydrodynamic output of EFDC was linked to the most recent, publicly released version of WASP (Water Quality Analysis Simulation Program (WASP),

2022, v. 8.4). This work combines discrete depth profile DO data along with continuous samples of DO (approximately every 15 min) and [chl *a*] to provide insight into how these different data types affect our perspective of the system as well as model performance. Then, a series of scenarios were simulated to investigate how loads of nitrogen from different sources influence DO at different locations and different depths over the year. These scenarios investigated the roles of atmospheric deposition, tributary loadings and WWTPs, and the role of sediments as fluxes of nutrients and sediment oxygen demand.

This study serves to (1) develop and apply a 3-D mechanistic water quality model for to improve understanding of hypoxia in Narragansett Bay, (2) investigate how well WAPS8 performs, (3) identify possible areas where mechanistic improvement could be made and where additional research would improve understanding, and (4) investigate how different sources of nutrients affect spatial and temporal resolution of hypoxia.

## 2. Methods

### 2.1. Site Description

Narragansett Bay is the largest estuary in New England (northeastern US) (Figure 1), primarily located in Rhode Island (RI) with a small northeast portion in Massachusetts (MA). The Bay has a surface area of 380 km$^2$, mean depth of 8 m, volume of $3 \times 10^9$ m$^3$, tidal range of 1 to 2 m, mean residence time of 26 d (range 1.67 d to 42.5 d), and annual freshwater input of $5 \times 10^6$ m$^3$ yr$^{-1}$ [26–28]. The Bay drains a 4836 km$^2$ watershed, with 40% in RI and 60% in MA, encompassing approximately 100 cities and towns and a population of over two million. The watershed is more urban than other New England watersheds that drain into the North Atlantic, with 38% of the land-use classified as urban, 40% forest, 12% wetland, and 5% agriculture [17]. Urban development surrounds the Bay, and Rhode Island is the second most densely populated (1018 persons per sq. mi. in 2010) state in the US [29], primarily bordering the Upper Bay near Providence (Figure 1). There are 37 WWTPs in the watershed, with 11 WWTPs releasing directly into the Bay [17]. Approximately $2.05 \times 10^6$ kg N yr$^{-1}$ enters the Bay, with approximately 80% of that coming from WWTPs [30]. The Bay is a complex network of water bodies with over 30 islands. The three largest islands, Aquidneck, Conanicut, and Prudence Islands, divide the bay into the West Passage, the East Passage, and the Sakonnet River. In the northwest, near Providence, RI, the Seekonk River flows into the Providence River, which flows into the Upper Bay. In the northeast, the Taunton River flows into Mount Hope Bay. In the west, there is a small side embayment, Greenwich Bay. Below the Upper Bay are the East and West Passages. To the east, there is a narrow section connecting Mount Hope Bay to the Sakonnet River. The East and West Pass and the Sakonnet River connect to the Rhode Island Sound [31].

Concentrations of annual total nitrogen [TN], total phosphorous [TP], ammonia [NH$_3$], phosphate [PO$_4$], and nitrite/nitrate [NO$_2$ + NO$_3$] decrease exponentially along the north to south axis of the Bay, from the Providence River to the mouth of the Bay, for two measured time periods, 1997–1998 and 2014 [32]. Summer surface [chl *a*] decreased along the north to south axis along Providence River, Upper Bay, mid-Bay, and Lower Bay in 1980/1997, 2006–2011, and 2013–2015. At the Providence River, mean [chl *a*] was 22 µg L$^{-1}$ and in the Lower Bay, mean [chl *a*] was 5 µg L$^{-1}$ [32]. Providence River, Upper Bay, Greenwich Bay, and part of Mount Hope Bay have been designated hypoxic zones [18].

### 2.2. Modeling Framework and Model Domain

The modeling effort incorporated the use of two modeling frameworks: Environmental Fluid Dynamics Code (EFDC) [33] and Water Quality Analysis Simulation Program (WASP8, v 8.4). Both EFDC and WASP have been presented by the US Environmental Protection Agency's (EPA) Office of Water as tools useful for developing management strategies (e.g., TMDL (Total Maximum Daily Load)) [34]. The Narragansett Bay EFDC model simulated hydrodynamics of the Bay for one year, 2009, using a grid of 754 horizontal segments and 8 vertical layers, totaling 6032 segments [35]. Dimensions of the segments vary across the domain, with an average width of 642 m (east–west) and an average length

of 1218 m (north–south). The thickness of each layer is location dependent, as the system divides the entire depth at a location by eight such that the thickness of each segment in a column is the same. As the volume of the entire column changes (as flow increases or as the tide comes in and out), each layer changes so that all layers remain uniform. WASP uses the output and model domain of EFDC, removing 3 rows at the seaward boundary (8 layers, 661 segments per layer, total of 5288 segments) as was done with a previous modeling effort using WASP7 (v7.52) [28] (Figure 2).

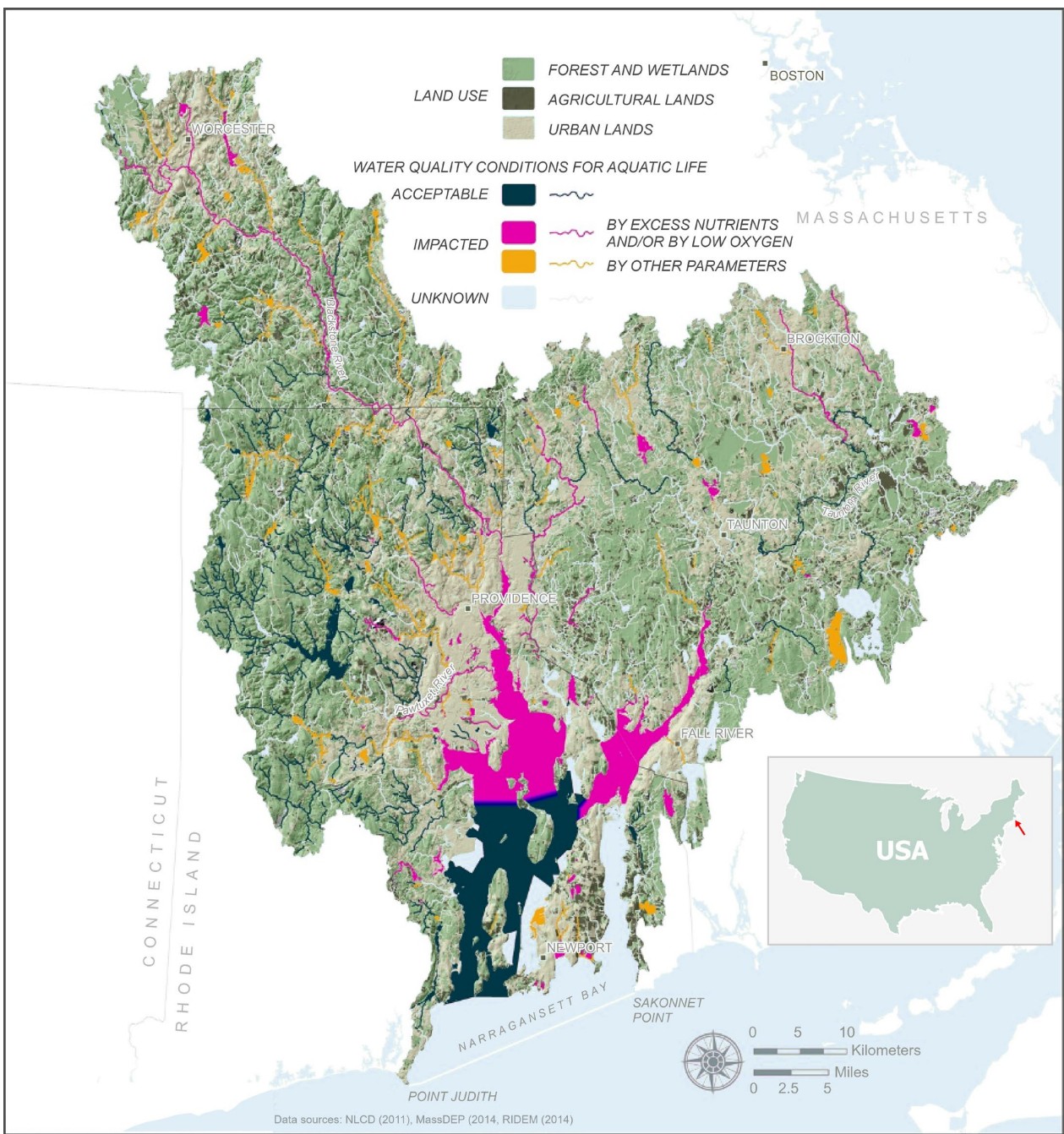

**Figure 1.** Site location of Narragansett Bay featuring details of land-use (forest and wetlands, agricultural, and urban) and locations of acceptable and impacted water quality (Adapted with permission from [17]).

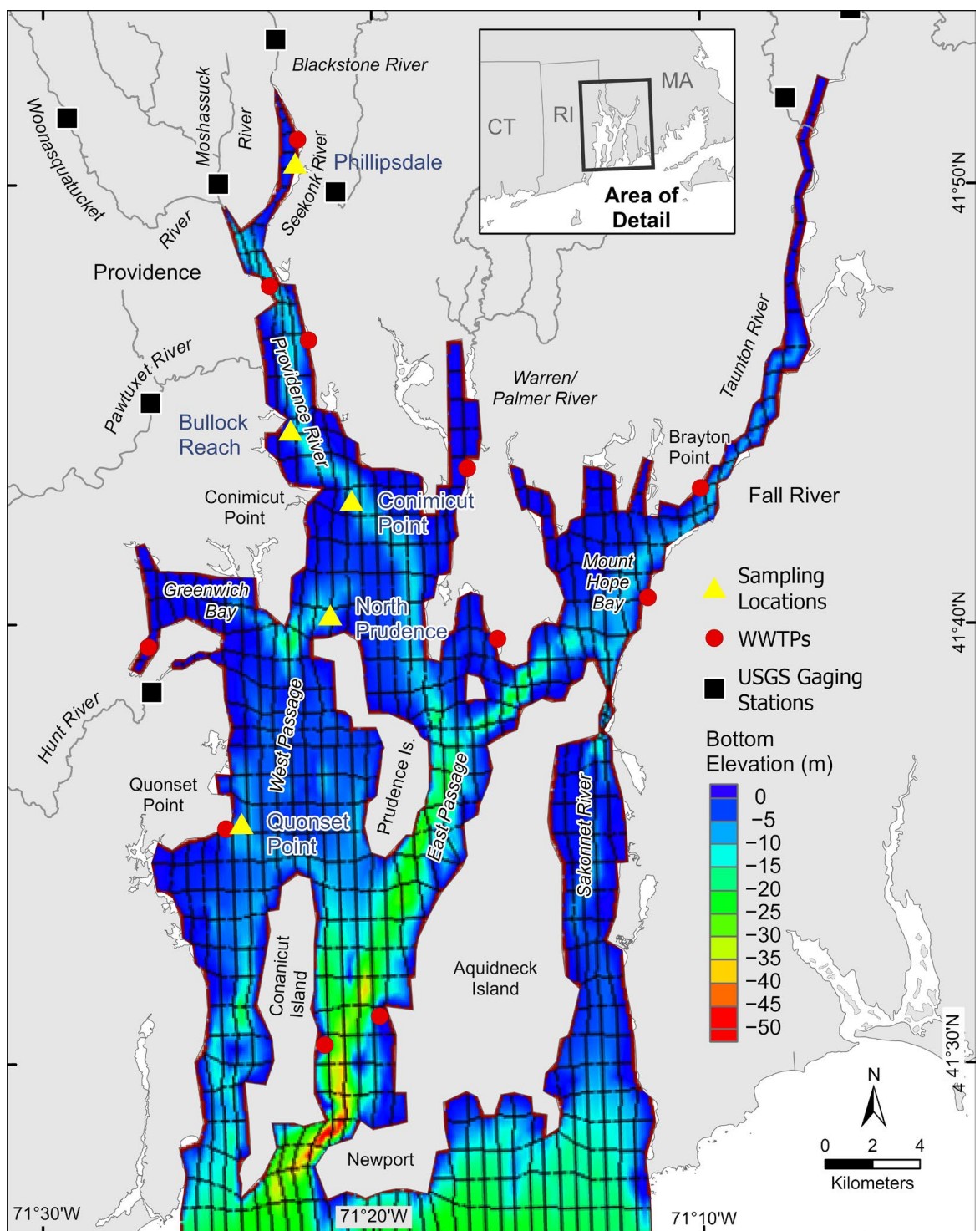

**Figure 2.** Narragansett Bay WASP Model domain. with bottom elevation (adapted [35]). Sampling locations (solid yellow triangle) for dissolved oxygen and chlorophyll *a*, from north to south: Phillipsdale, Bullock Reach, Conimicut Point, North Prudence, Quonset Point (adapted [28]). Wastewater Treatment Plants (WWTPs) are marked by red disks and tributary inflows are marked by solid black squares [35,36].

The Environmental Fluid Dynamics Code simulates hydrodynamics in multiple dimensions (https://www.epa.gov/ceam/environmental-fluid-dynamics-code-efdc (accessed on 7 September 2018) [33,37]). The model has evolved over the decades and is one of the most widely used hydrodynamic models in the world. It uses stretched (sigma) vertical

coordinates and curvilinear, orthogonal horizontal coordinates to define the model domain. EFDC solves dynamically coupled equations for turbulent kinetic energy, turbulent length scale, salinity, and temperature. Further details on the calibration and results of EFDC are described elsewhere [35,36]. The simulated EFDC results create a hydrodynamic file (*.hyd), which is imported into WASP (*.wif). Specifically, WASP imports water inflow from tributaries and other loading sources (e.g., WWTPs), tidal exchanges, advection flow, dispersion exchange, water temperature, and salinity. The time step for EFDC simulation was 120 s.

The Water Quality Analysis Simulation Program (WASP) is a dynamic, compartment modeling framework capable of simulating aquatic ecosystems in multiple dimensions. WASP allows the user to develop site-specific models to simulate concentrations of conventional pollutants (Advanced Eutrophication Module) and toxic materials (Advanced Toxicant Module) in the water column and sediment layers (https://www.epa.gov/ceam/water-quality-analysis-simulation-program-wasp (accessed on 10 February 2021)) [38,39]. WASP is one of the most widely used water quality models in the US and the world and has been widely applied for management practices and regulatory action. WASP has gone through continual advances and versions over the past 50 years, and the current version is WASP8 (version 8.4).

### 2.3. WASP Model State Variables and Governing Processes

WASP is a modeling framework where the user selects the state variables to be simulated, their governing processes, and the parameters and constants governing those processes. In some instances, there are multiple options for state variables and their processes to choose from. In the WASP Advanced Toxicant module, the user defines state variables (e.g., benzene, graphene oxide, silt) from a range of state variable classes (e.g., chemical solute, nanomaterial, solid) [39]. The Advanced Eutrophication module, used here, specifically defines state variables (e.g., nitrate, ammonia, dissolved organic phosphorous). For the Narragansett Bay WASP model, 15 state variables were selected and simulated (see Table 1), capturing the full extent of WASP's DO, nutrient cycling, and phytoplankton dynamics. Two different types of ultimate Carbonaceous Biochemical Oxygen Demand (CBODU) were simulated, carbonaceous biochemical oxygen demand and as BOD formed by phytoplankton death and decay, to allow for their different decay rates.

**Table 1.** System state variables.

| WASP System Type | System Name | Units | Initial Concentration |
|---|---|---|---|
| $NH_3$ | Ammonia | mg N $L^{-1}$ | 0.011 |
| $NO_3/NO_2$ | Nitrate/Nitrite | mg N $L^{-1}$ | 0.035 |
| ORG-N | Dissolved Organic Nitrogen | mg N $L^{-1}$ | 0.048 |
| D-DIP | Inorganic Phosphorous | mg P $L^{-1}$ | 0.027 |
| ORG-P | Dissolved Organic Phosphorous | mg P $L^{-1}$ | 0 |
| CBODU | Ultimate Carbonaceous Biochemical Oxygen Demand ($CBOD_U$) | mg CBOD $L^{-1}$ | 0 |
| CBODU | Ultimate Biotic Biochemical Oxygen Demand ($BOD_U$) | mg BOD $L^{-1}$ | 0 |
| DISOX | Dissolved Oxygen | mg $O_2$ $L^{-1}$ | 10.14 |
| DET-C | Detrital Carbon | mg C $L^{-1}$ | 0.014 |
| DET-N | Detrital Nitrogen | mg N $L^{-1}$ | 0.0034 |
| DET-P | Detrital Phosphorous | mg P $L^{-1}$ | 0.00079 |
| TOTDE | Total Detritus | mg $L^{-1}$ | 0.044 |
| SALIN | Salinity | PSU | 0 |
| SOLID | Total Suspended Solids | mg $L^{-1}$ | 0 |
| PHYTO | Phytoplankton | µg chl *a* $L^{-1}$ | 0.3 |

Figure 3 presents the nutrient cycling processes for all the state variables simulated in the modeling framework for each WASP segment. Many constants and parameters are used in the equations being solved by WASP for every segment at every time step. Details on the equations are presented elsewhere (WASP documentation, lecture notes, and user's guides (available at https://www.epa.gov/ceam/water-quality-analysis-simulation-program-wasp (accessed on 14 September 2020), http://epawasp.twool.com/ (accessed on 14 September 2020)). Parameters used in the simulations, grouped by the type of state variable for which they are used, are provided in the Appendix A.

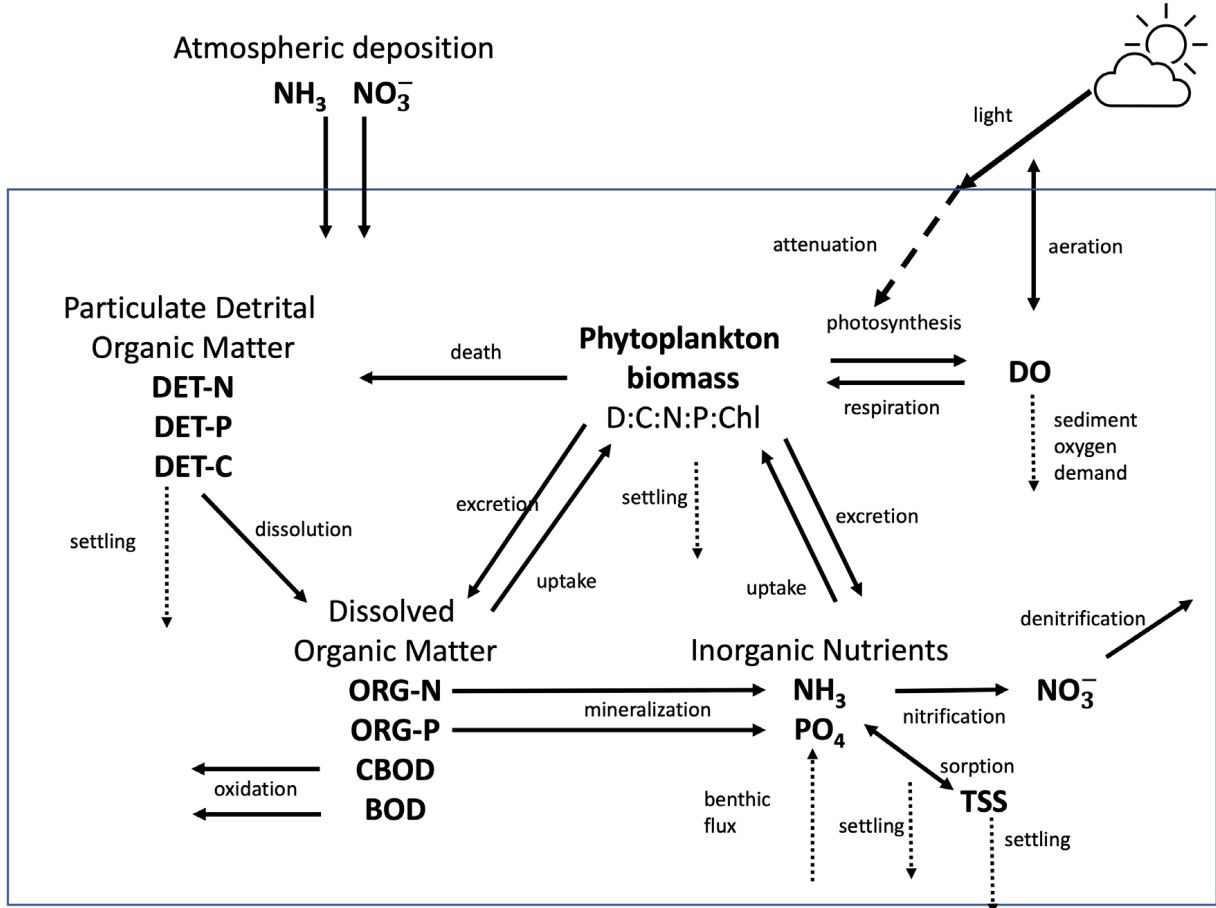

**Figure 3.** Nutrient Cycling Processes and Simulated State Variables in Narragansett Bay WASP Model. The 15 state variables are presented in bold: nitrate/nitrate (represented as nitrate, $NO_3^-$), ammonia ($NH_3$), inorganic phosphorous ($PO_4$), detrital-N, detrital-P, detrital-C, organic-N, organic-P, CBOD, BOD, dissolved oxygen (DO), total suspended solids (TSS), and phytoplankton. Processes are represented with arrows showing direction of the process. Dotted lines represent state variables interaction with the sediments, including settling, sediment oxygen demand, and benthic flux.

### 2.4. Model Application

The Narragansett Bay WASP model incorporates a range of model inputs, including tributary inflows, loads, and other forcing functions. Specific values, time series, and details are provided in the Supplementary Materials. The Narragansett Bay WASP model is applied for a series of different scenarios. First, the model is applied to the current condition of 2009, which allowed for model calibration, evaluation, and perspective of DO and phytoplankton in the Bay. Observations were taken at Phillipsdale, Bullock Reach, Conimicut point, North Providence, and Quonset Point (Figure 2). These locations were chosen because they capture the transect from the top of the Bay in the Seekonk River, downstream of a WWTP, along the Providence River, down the West Passage. Model

results were evaluated using $R^2$, Root Mean Square Error (RMSE), and Nash–Sutcliffe (NS) metrics (data processing details in Supplementary Materials). The focus on this effort was on simulating DO, since that is the water quality constituent of concern in the Bay, with [chl *a*] also being used for model evaluation. Nutrient data in the Bay are sparse and were not adequate for calibration or evaluation. WASP internally calculates its time step optimized for numerical stability, with a minimum time step of 0.0001 d and model output every hour. Upon having a calibrated model for 2009 ("Base Case"), a series of scenarios were simulated to investigate how loads of nitrogen from different sources influence DO concentrations at different locations and different depths over the year. These scenarios investigated the roles of atmospheric deposition ("No Deposition"), tributary loadings and WWTPs ("No Boundary"), and the role of the benthic fluxes of nutrients and sediment oxygen demand ("No Sediment Input"). For these scenarios, the input was set to zero. Each of these scenarios was a separate model simulation. Then, all scenarios were combined to form one simulation with all these sources turned off ("No Inputs"). Because the Bay operates as an open system and is non-linear in responses, performing all these simulations plus the combination of all of them allows us to mechanistically investigate what is governing DO in different locations with depth due to different governing processes. While these scenarios would not be realistic, they allow us to use the model in a way to effectively perform experiments to provide deeper mechanistic understanding of the system.

## 3. Results

### 3.1. Dissolved Oxygen and Phytoplankton Simulation

The 2009 simulation results for DO are presented for five locations from north to south, including Phillipsdale, Bullock Reach, Conimicut Point, North Prudence, and Quonset Point (see Figure 2), as columns from left to right (Figure 4). The rows represent the different layers simulated, with the surface as the top row and the bottom layer as the bottom row. The two types of observations are presented as red dots for discrete depth profiles and the green lines are the continuous (15 min interval) sonde data. Depth profile data were only available for Phillipsdale, Bullock Reach, and Conimicut Point. Sonde data were available for all locations but only at certain depths (surface and deep layers).

For all locations, the simulations follow the annual sinusoidal pattern of DO solubility, with higher concentrations in the colder months of winter and lower concentrations in the warmer months of summer. The top layer hovers around solubility due to reaeration with the overlying air. Going down the rows with increasing depths, the simulated concentrations drop lower in the summer months. With depths, the model captures the variability in DO with increasing DO during the day due to phytoplankton growth and then decreasing DO as phytoplankton respires. Summer DO drop closer to zero and become hypoxic. The deepest layer (row 8) shows DO reaching anoxia during summer months.

The simulated DO does well at capturing the discrete DO depth profiles. For the sonde continuous data DO, the simulations capture the general trend but do not capture the large swings of daily DO. This is a much larger issue in the surface layers than the deep layers. For the deep layers, the simulated DO does a much better job of capturing the low ends of DO. This is reflected in the deep layer sonde data for Bullock Reach, Conimicut Point, and Quonset Point. There is less observed daily variation of DO for these locations, and the model captures the low DO in late summer. For Phillipsdale, the DO has a much wider variability similar to the surface layers. The water at Phillipsdale, averaging approximately 2 m, is shallow compared to the rest of the Bay, which may explain the higher variability of DO compared to deeper segments closer to the ocean boundary. For management needs, it may be less important to capture the increases and swings in DO than to capture the times and locations of low DO. This is particularly the case for anoxia and hypoxia, which the model does well capturing in the deep layers.

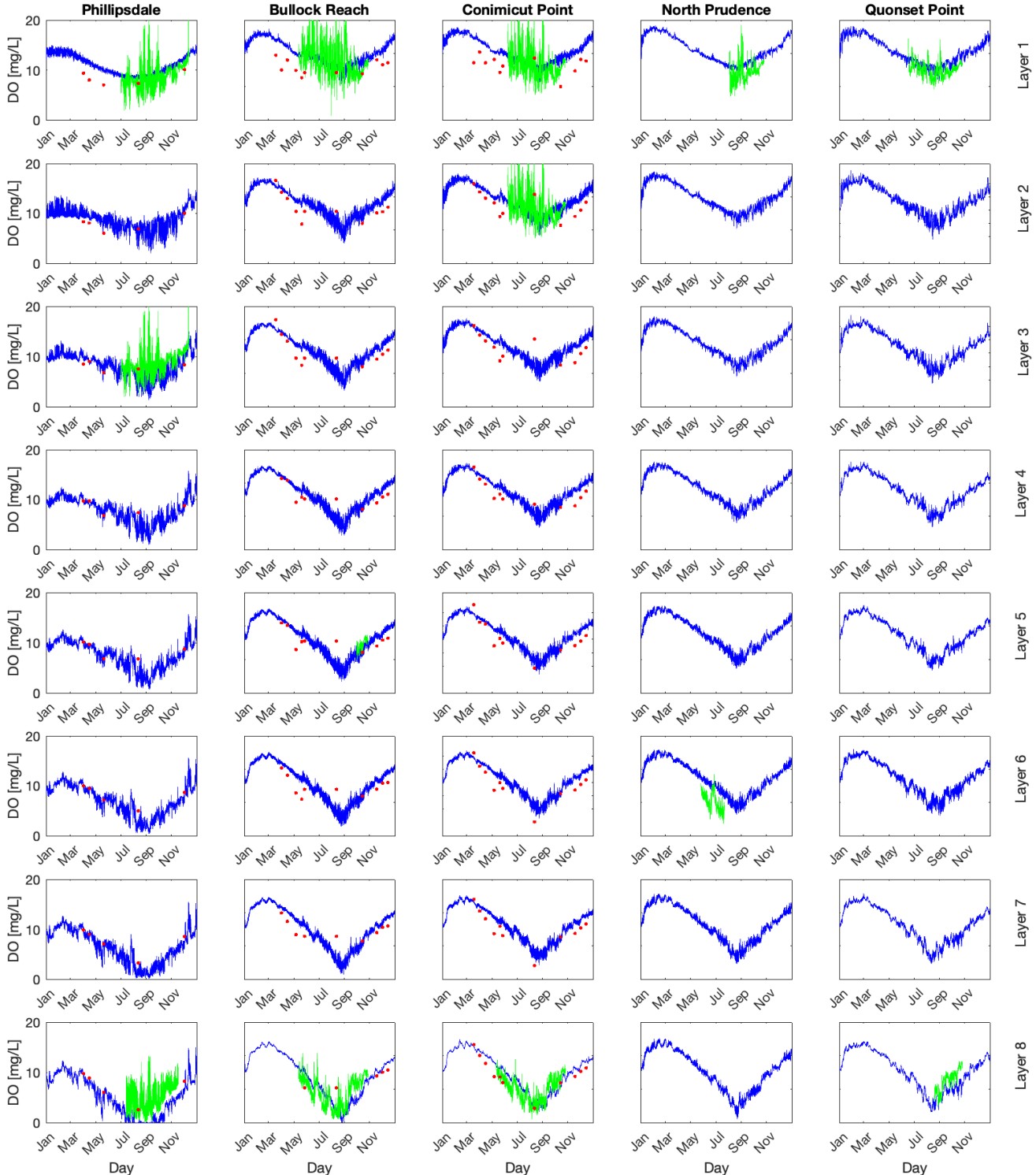

**Figure 4.** Dissolved Oxygen Concentrations [mg/L] in Narragansett Bay at 5 locations with depth. Each column represents where there is observed data, from north to south on the west side of the Bay. Each row represents a water column layer, from Layer 1 at the surface to Layer 8 right above the sediments. The solid blue lines represent the WASP simulated DO. The solid green lines are the observed sonde data (15 min intervals). The red dots are the depth profile samples, only available at the first three locations.

The simulations of phytoplankton are shown in Figure 5 with the same layout of rows and columns and locations as Figure 4. Simulated [chl *a*] is presented in blue, and the continuous (15 min) sonde observations in green. Depth profile observations were not available for chl *a*. The simulations demonstrated the bloom in spring and the die off in autumn throughout the system, with decreasing phytoplankton concentrations from north to south (left to right column) as well as decreasing from surface to depth. Phillipsdale demonstrated a bloom in spring with a die off followed by a second bloom in the fall, which matches the observed pattern in the third layer. Comparing to observations, the observations had much higher swings from high [chl *a*] to low [chl *a*] than the model predicted. The model predicted daily swings of [chl *a*], with [chl *a*] growing in the spring, maintaining an elevated concentration through summer and decreasing towards zero in autumn.

Calibration and parameter adjustment was unable to capture the large diurnal variability in either [chl *a*] or DO. For example, increasing phytoplankton growth or respiration rates results in nutrient depletion, preventing further growth. Changing phytoplankton stoichiometry (e.g., C: chl *a*) results in shifting the mean [chl *a*] without sufficiently increasing the range.

### 3.2. Model Evaluation: DO and Chl a

Simulated versus observed for continuous sonde data for DO and [chl *a*] where sonde data are available are presented in Figure 6 and discrete depth sample DO for all layers plus a composite for all depths for Phillipsdale, Bullock Reach, and Conimicut Point in Figure 7. The bottom row for Figure 7 combines all depths.

The model does relatively well capturing the depth profile data (Figure 7). The composite $R^2$ for the last row suggests that overall, the model is doing well for Phillipsdale and Conimicut Point, while the model has more challenges with Bullock Reach. Sonde observations have large ranges over short time periods, which the model does not capture. The disconnect between the sonde (continuous) and depth profile (discrete) comparisons for model evaluation presents an interesting challenge. The sonde data observe large ranges over a given day for both DO and [chl *a*], while the depth profile observations fall relatively in line with the simulated results. This is particularly interesting that the discrete, depth profile samples do relatively well for all layers for the three locations where there are data. In previous studies, grab samples were the only available data (see, e.g., [23]). The advent of continuous sampling provides additional insight on understanding DO dynamics and how well the model processes are capturing those changes. Sonde data provide a high temporal resolution (15 min) of data at a very specific location (a single WASP layer). The depth profile samples have less temporal resolution but provide information over the entire water column (every WASP layer at a location). Comparing the values of the depth profile data to the sonde, the model simulates the depth profile samples well, which never exhibit a spike or drop that the sonde data observed. However, it is possible that the samples were not at the timing of the large swings. It is also possible that these swings in concentrations may be a challenge with using sonde data, as there may be errors in the observations. If these are large swings in DO and [chl *a*], then these observations suggest that there may be highly dynamic processes of phytoplankton growth and respiration that the model cannot capture with its current model structure or parameterization. The model simulates average concentrations across an entire segment volume, which may not capture large variations in a specific location over the 15 min intervals. These issues are important to recognize both for improving process representation in the model design as well as evaluating model performance.

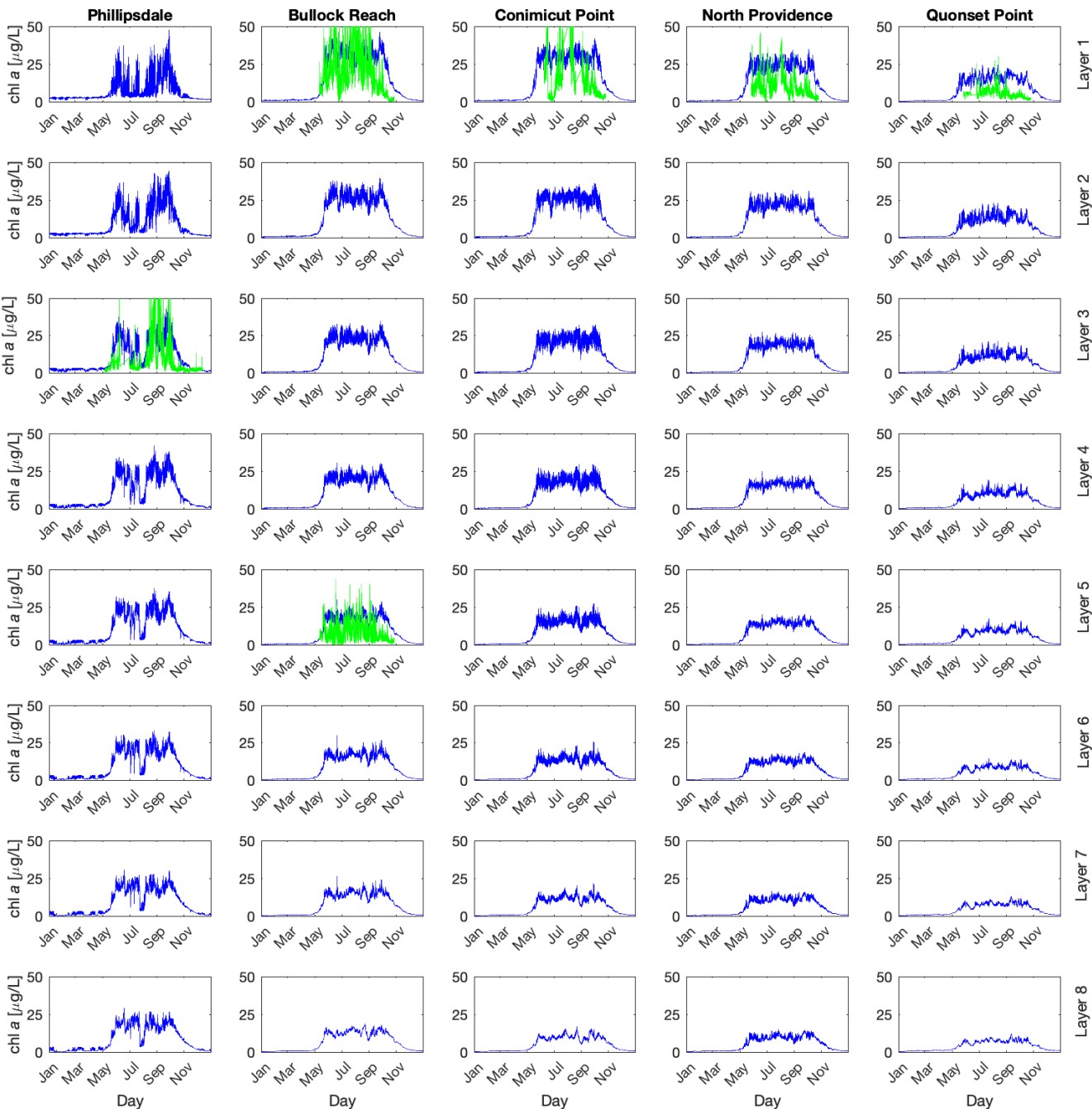

**Figure 5.** Chlorophyll *a* concentrations in μg L$^{-1}$ in Narragansett Bay at 5 locations with depth. Each column represents where there is observed data, from north to south along the West Passage. Each row represents a water column layer, from Layer 1 at the surface to Layer 8 right above the sediments. The solid blue lines represent the WASP simulation output. The solid green lines are the observed sonde data.

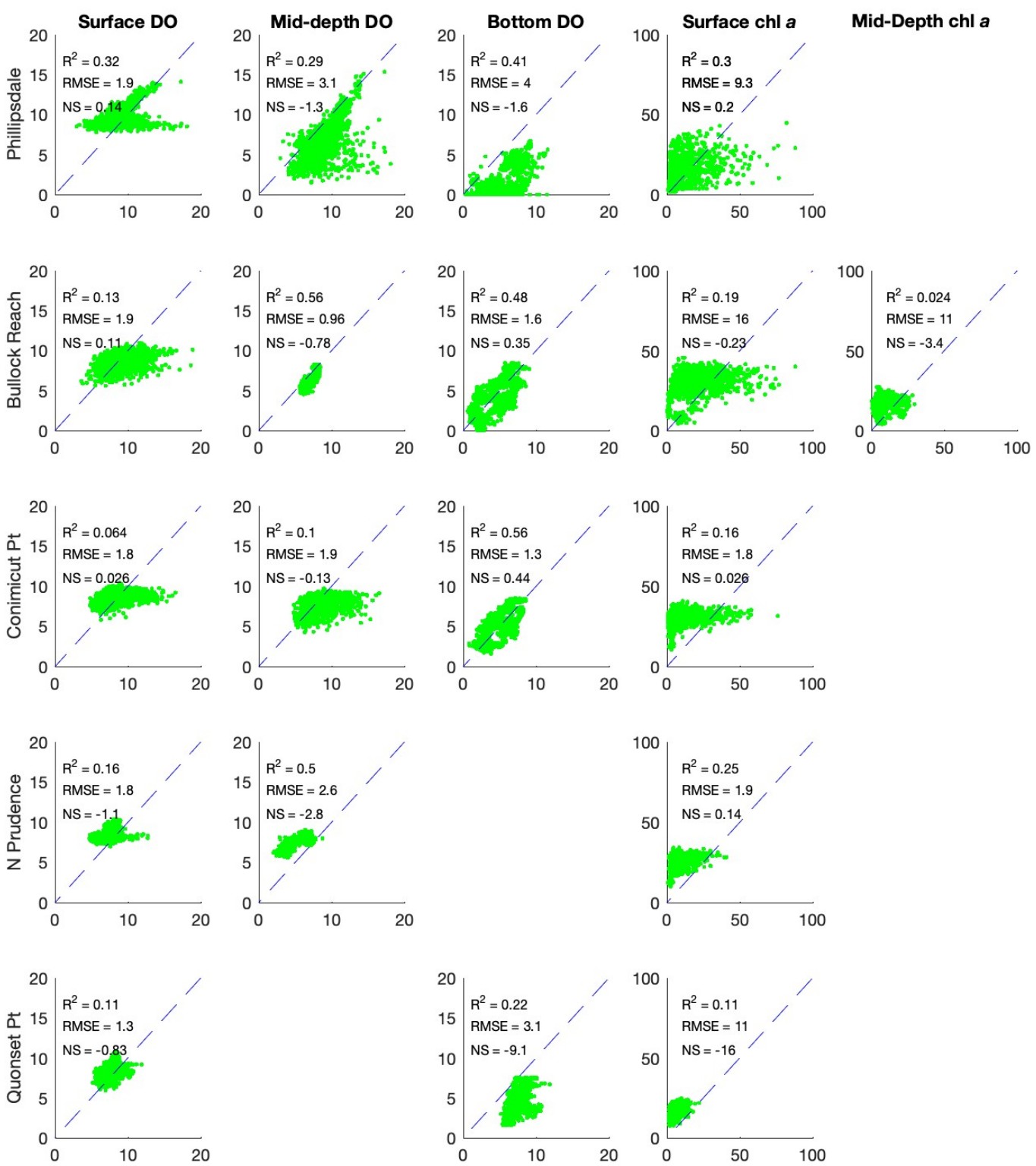

**Figure 6.** Simulated WASP vs. Observed Continuous Sonde Dissolved Oxygen and Phytoplankton Concentrations with Calculated R$^2$, Root Mean Square Error (RMSE), and Nash–Sutcliffe (NS). Simulated WASP vs. Observed Sonde data. First 3 columns are DO at Surface, Mid-depth, and Bottom. Columns 4 and 5 are chl *a* at surface and mid-depth. Rows are locations from north to south.

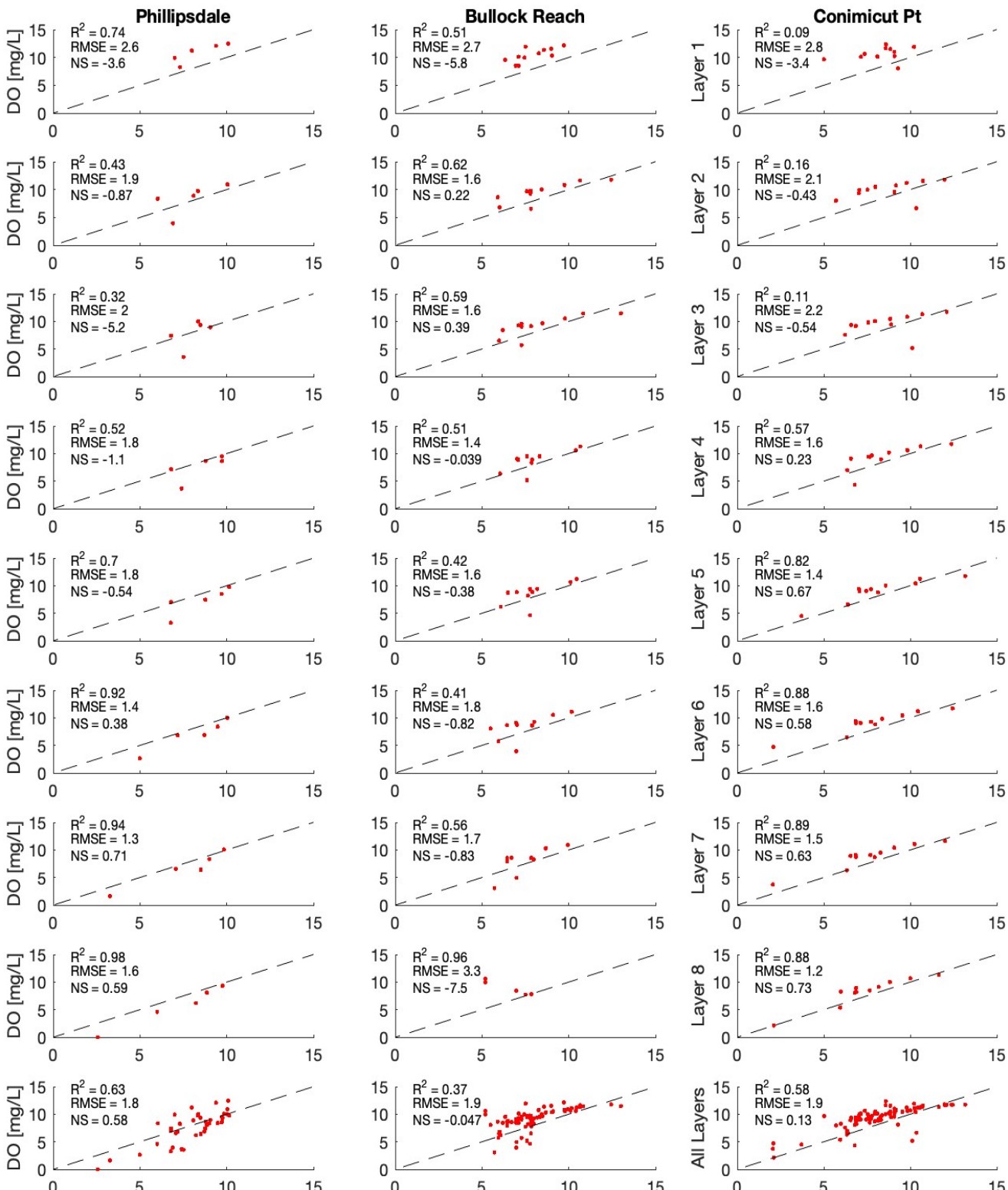

**Figure 7.** Simulated WASP vs. Observed Discrete Depth Profile Dissolved Oxygen Concentrations with Calculated R$^2$, Root Mean Square Error (RMSE), and Nash–Sutcliffe (NS). Simulated WASP vs. Observed Sonde data. First 3 columns are DO at Surface, Mid-depth, and Bottom. Columns 4 and 5 are chl *a* at surface and mid-depth. Rows are locations from north to south. Simulated WASP vs. Grab Samples for DO. Columns are 3 locations with available depth profile DO data. First 8 rows are water column layer, row 9 is the combining all layers of data.

### 3.3. Dissolved Oxygen Heat Maps

Figure 8 presents simulated DO for 2009 with depth over the course of the simulation year, from upstream (Phillipsdale) to downstream (Quonset Pt). Hypoxic waters are shown in black to dark red, with oxic waters being yellow to white. Figure 8 illustrates some of the utility of using a mechanistic model for simulating DO; the model provides simulated DO for all layers at all times at across the extent of the Bay. The zones and time periods of oxic waters are clear, and the rise and spread of anoxic and hypoxic waters are well captured by the appearance and growth of the black and deep red colored zones. Similarly, the model output shows the difference between the upstream zones at Phillipsdale and Bullock Reach and the hypoxic waters throughout summer and the decreasing zone of hypoxic waters going downstream. Phillipsdale has hypoxic waters develop early in Spring, unlike the other locations. For Phillipsdale, the hypoxia develops and slowly penetrates into the upper waters. The upper layer remains oxic throughout the entire year, demonstrating the impact of aeration in the upper layers. These figures show the transitions as the DO starts decreasing the rise of hypoxic waters with depth and then the return to more oxic waters in autumn. These figures also show how the hypoxia lasts longest at Phillipsdale and how it takes until late autumn for the oxic waters to penetrate all layers.

### 3.4. Model Scenarios for Nitrogen Controls on Dissolved Oxygen

Different model case scenarios were constructed and run to investigate how different loads of nutrients to the Bay were controlling DO. One of the important aspects of this work pertains to understanding what is controlling the hypoxic and anoxic zones so that management strategies could be developed to address them. To that end, using the "Base Case" Model developed for 2009 as discussed earlier, four other models were developed. These included "No Boundary", "No Deposition", "No Sediment Input", and "No Inputs". The "No Boundary" model had tributary concentrations of 0 mg/L for ammonia and nitrate/nitrite. The "No Deposition" model set atmospheric deposition of nitrogen to zero. The "No Sediment Input", turned off sediment oxygen demand (SOD), benthic ammonia flux, and benthic phosphate flux. The "No Inputs" model combined all three of the new cases.

Figure 8 presents the simulated results of DO for surface (layer 1), mid-depth (layer 4), and deep (layer 8) water by column, and each row is for each location. The five different model cases, including the "Base Case", are plotted on each sub-plot so that they can be readily compared. The process equations governing DO as it relates to other state variables in the system is non-linear, as is demonstrated by the figures. For example, in some cases removing a nutrient source increases DO and in some instances it reduces DO. For all the different cases, the surface DO is minimally impacted by removal of the nutrient source. This is most likely due to the reaeration process in surface layer, buffering the impact on DO. Surface DO in Phillipsdale is minimally impacted. Surface DO at Bullock Reach shows that removing the tributary inputs appears to reduce DO during the summer, while removing sediment influence results in increased DO during the summer. Surface DO in Conimicut Point shows that reducing boundary inputs results in increased DO. For North Prudence and Quonset Point, surface DO appears relatively unchanged across the model options.

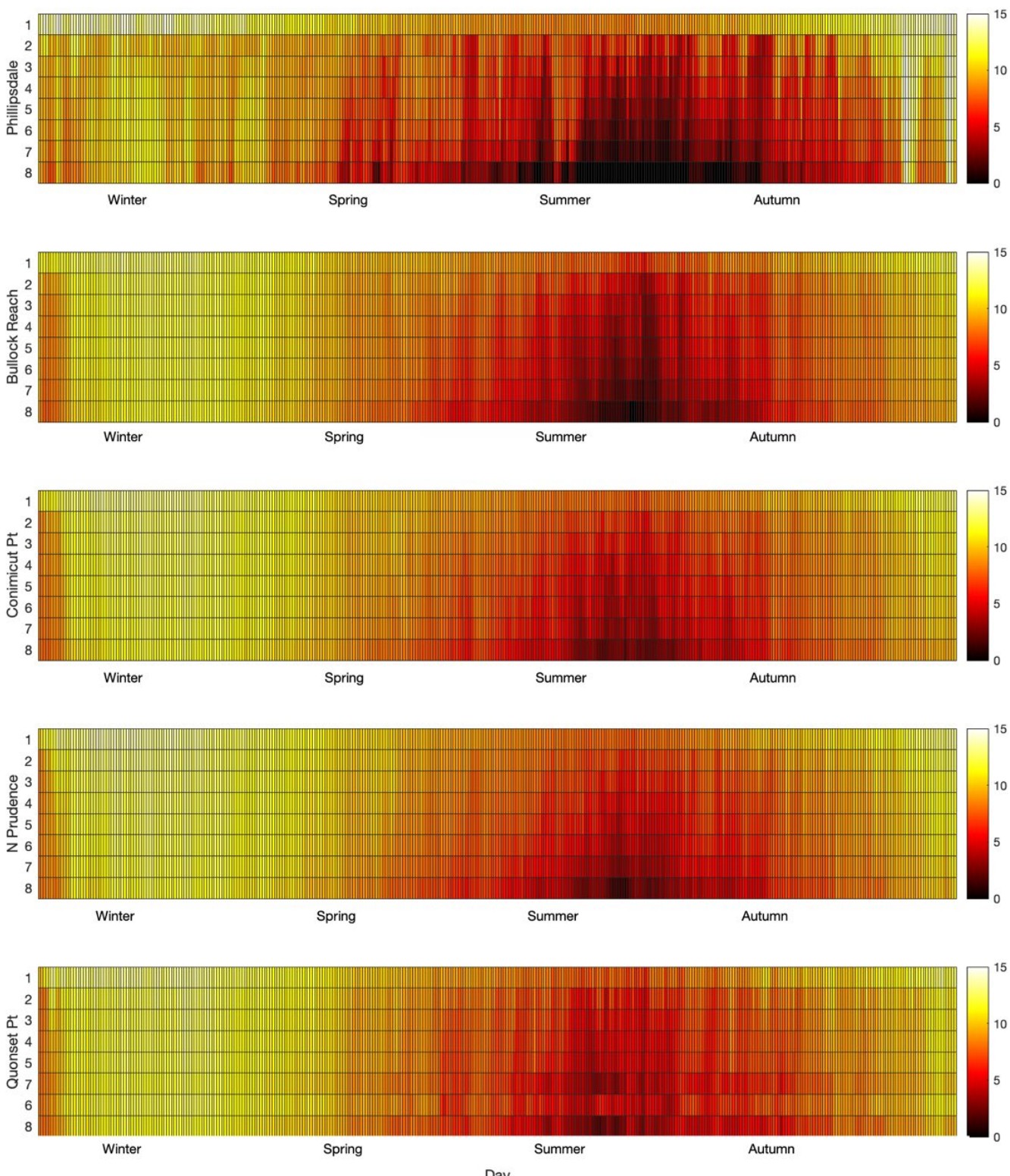

**Figure 8.** Simulated dissolved oxygen concentrations with time (day) on the x-axis and depth as layer on the y-axis for five locations from north to south. Each square represents the DO for a given day in that layer on that day at the location. DO ranges from 0 mg L$^{-1}$ (black) to 15 mg L$^{-1}$ (white).

For the mid-depth and deep layer DO, there is much more variability due to the different model scenarios. For all of scenarios, the removal of sediment influence increases DO in the mid-depth and deep waters for all locations. For Phillipsdale, there is a wide

variability in DO when sediment influence is removed and DO remains much higher than other cases. For all cases, the "No Inputs" and "No Sediment" have similar responses for all locations for mid-depth and deep layers. The "No Sediment" shows much more variability than the "No Inputs", suggesting how the presence of incoming nutrients influences the growth and respiration of phytoplankton and the associated rise and fall of DO. The "No Deposition" case shows minimal difference than the "Base Case", suggesting that atmospheric deposition of nitrogen may not be an important driver for DO in the Bay. The "No Boundary" also does not have an appreciable impact on DO in this system, suggesting that the current loading of nutrients may not be governing the lower DO in the mid-depth and deep layers. However, the "No Sediment Input" model results are different than the "No Inputs", suggesting that if the sediment influence is removed, the influence of the tributary loads may become more important. This analysis suggests that the sediment nutrient flux and SOD may be the primary driver for low DO in the mid to deep waters, yet the nutrient loads in the tributaries would become more important as sediment influence decreases. Without sediment nutrient fluxes, water clarity could improve allowing for increased light penetration in the water column, potentially increasing photosynthesis and DO. The other important aspect of this analysis is that because this model is simulating a single year, the sediment flux is held constant at observed values. As nutrient loads change in the Bay, the sediment would correspondingly respond and change. In our current model, the sediment flux and SOD are fixed, which is appropriate for the single year simulation.

## 4. Discussion

The observations and WASP model simulations show the rise and fall of DO for the five sites of interest over the course of the year. The upper reaches (Phillipsdale and Bullock Reach) are typically stratified, with density driven circulation patterns, the middle section of the Bay (Conimicut Point and North Prudence) is weakly stratified or mixed, and the lower Bay (Quonset Point) is well-mixed [26]. The DO and phytoplankton observations and simulations reflect this structure (Figures 4, 5 and 8). Farthest upstream (Phillipsdale), hypoxia develops near the sediments early in the spring and then moves up the water column during summer until returning in autumn. Additionally, phytoplankton blooms (both observed and simulated) occurred in spring and then autumn. These results align with hypoxia driven by tributary nutrient loads [40]. Bullock Reach hypoxia begins later than at Phillspdale and penetrates up towards the surface. Conimicut Point, North Prudence, and Quonset develop hypoxia later, with shorter durations and less penetration towards the surface.

Fennel and Testa proposed an approach for relating hypoxia timescale to residence time as a 1:1 ratio [41]. The upper reach of the Bay (Providence-Seekonk River) has a residence time of 0.8 d to 13 d [42], which compares to the Pearl River, China (4 d) and the East China Sea (11 d) [41]. The upper reach behaves similarly to both of these systems, exhibiting summer hypoxia driven by sediment oxygen demand [43]. Using the approximate 1:1 relationships hypoxia timescale to residence time metric, the simulations support that upper reaches hypoxia faster than farther downstream segments. Using their designations as well, this study suggests that the upper reaches are more river-dominated systems compared to the middle and lower Bay. The main body of the Narragansett Bay has a residence time of 26 d (1.67 to 42.5 d) [26,42], which compares to the Northern Gulf of Mexico (30 d), while the Baltic Sea has a much higher residence time (3100 d) [41]. In the Gulf of Mexico, the Mississippi River delivers large volume of freshwater carrying high nutrient loads, which results in a thin layer of hypoxia near the sediments [44]. The Baltic Sea exhibits permanent stratification with large zones of hypoxia [45]. Narragansett Bay is different from these system in that it does not exhibit strong or permanent stratification; the thickness of hypoxia is location and time dependent (Figure 8). The estimated small hypoxia timescale supports the quick rise and decline of hypoxia in the Bay, which may suggest management strategies to reduce nutrients released into the Bay may have a faster

response for recovery. Recent research has suggested that improvements to WWTPs have resulted in decreased hypoxia, nutrient concentrations, and phytoplankton growth [19,32].

The effect of wind can disrupt hypoxia, as seen in the Gulf of Mexico where high wind events mix the water column. Hypoxia in the Gulf is reestablished quickly when the wind-induced mixing subsides [44]. Recent modeling has also shown wind-driven hypoxia, where winds caused bottom water upwelling, pushing hypoxic waters to the nearshore [22]. The Bay is relatively protected from winds, and while strong wind events can impact flows in the passages, generally wind driven effects are minimal and serve mainly to promote mixing [26]. Some observations have suggested possible upwelling in Bullock Reach in the Providence River (personal communication, Narragansett Bay Estuary Program).

In this model, CBOD was simulated as two different parameters. Recent research has suggested that terrestrial CBOD may be subject to photochemical reactions, which can provide nutrients for microbial communities [46]. Currently, WASP does not incorporate this process, which could potentially account for increased phytoplankton growth. This could in turn account for the large diurnal variations in DO and phytoplankton concentrations, which the WASP model is not currently able to simulate.

The penetration of hypoxic waters from the deep waters into the water column suggests the importance of oxygen demand and nutrient fluxes from the sediment (Figures 8 and 9). The well-mixed and weakly stratified structure of the Bay supports this results. In the Baltic Sea, the role of sediments was found to change with DO levels in overlying waters, shifting from nitrogen removal to nitrogen release as hypoxia worsens [45]. As nutrient reduction management strategies are put into place, it is unclear what the response of the sediments will be and how fast changes may occur. The presence of legacy nutrients in the Bay may result in a lag in the water quality response and some zones of the Bay may change at different rates than other zones. Looking to long-term recovery of the Bay, it will be important to incorporate sediment diagenesis processes with adequate parameterization to investigate the effect of management strategies in the Bay watershed as well as implications for land-use and climate change.

Mechanistic modeling scenarios allowed for investigating the relative importance and impact of the role of different sources of nutrients and oxygen demand. The upper water column showed the importance of reaeration on governing DO and the influence of tributary sources of nitrogen including WWTP inputs, though the influence was small compared to the influence of deeper waters. The daily fluctuations in DO were driven by the growth and respiration of phytoplankton, while the overall shape of DO may be primarily driven by the sediment influence.

This research used two different types of observed data. The discrete depth profile data provided DO detail for each of the different WASP layers at specific times throughout the year. The sonde data provided high-resolution time DO and [chl *a*] at a very specific location over an extended time period. Oftentimes, discrete data are the only data available for model development and evaluation (e.g., [21–23]). The use of sonde data helps improve development and evaluation. WASP simulates large volumes of water as continually mixed reactors Given the large volume being simulated, WASP averages concentration over the entire volume, while the observed data are at a very specific location in that simulated volume. One area of future research could incorporate a more refined grid, with more cells of smaller volume, and/or more layers, all of which would have a cost of increased computational run-time as well as managing appreciably larger data sets to manage.

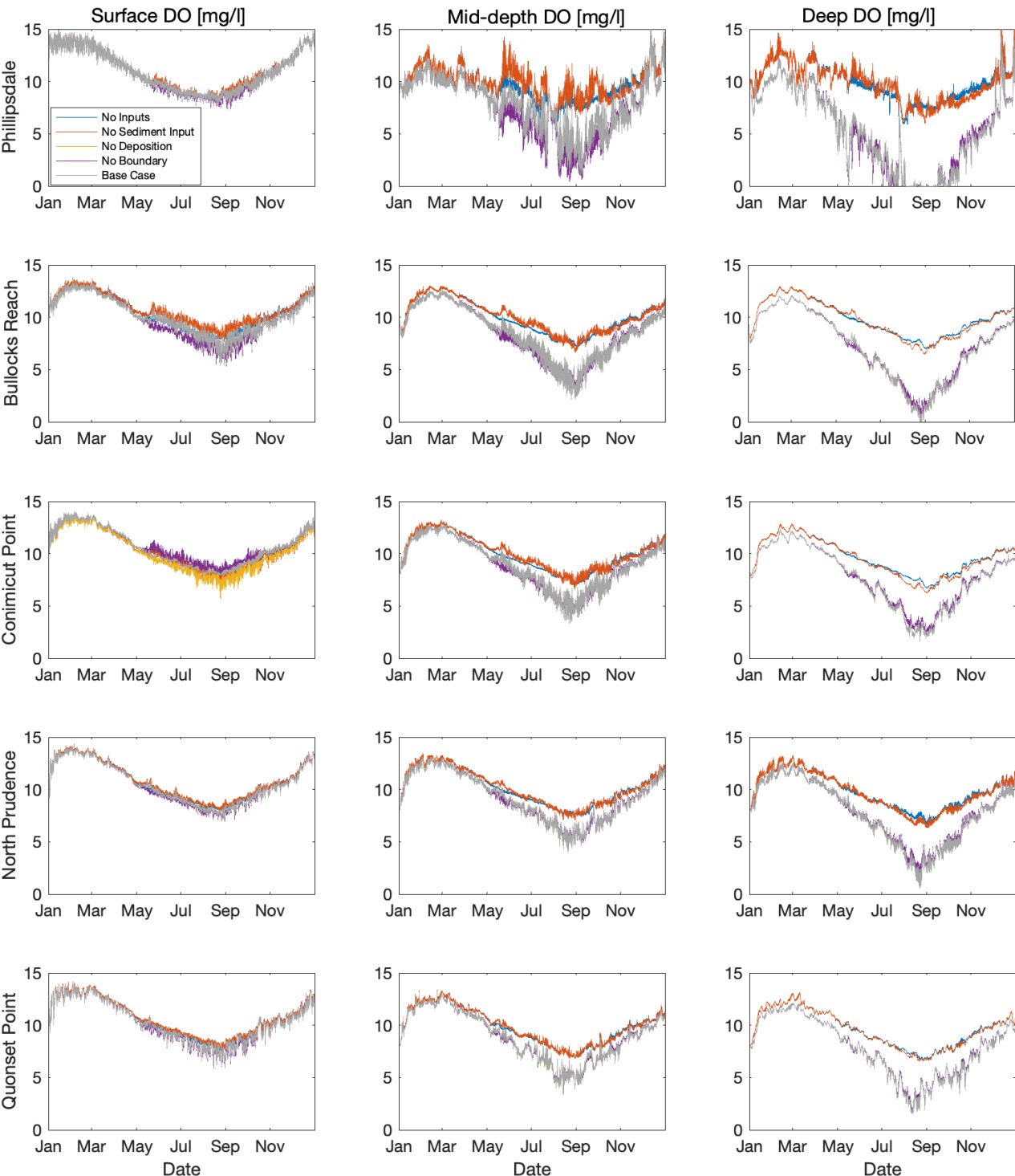

**Figure 9.** Dissolved oxygen WASP simulations over the year comparing different scenarios of nitrogen inputs. Columns are layer of interest and rows are the location of interest.

The use of the observed sonde data provides information into the processes going on at that scale and suggests that it is possible that DO and [chl *a*] may be covering a wider range than WASP is structured to simulate. The Narragansett Bay WASP model may be applicable to understand and capture general trends over longer time frames, but not refined down to the smaller time scales less than a day. These simulation results may be enough for management strategies on a larger scale and useful for looking at scenarios of management strategies. However, it does highlight that further research on

understanding how to represent these diurnal variations. Interestingly, the discrete samples were simulated quite well, which were the type of data previously available to previous estuary model applications. This creates an interesting challenge with doing simulation modelling when different types of observed data are providing different information on model performance. This also points to further research to investigate if our modeling framework is adequately capturing the necessary processes to a resolution necessary to make the appropriate and defensible management decisions, and how we might be able to simulate these finer-scale processes to improve our representations and simulations in the future.

## 5. Conclusions

A 3-dimensional mechanistic fate and transport model was developed for the year 2009 for the Narragansett Bay. The model helps inform our understanding on hypoxia and serves as a foundation for further studies, identifying scientific research needs as well as the potential for investigating longer time frames, climate change, LULC change, and acidification. The model was able to capture the trends and patterns of DO and phytoplankton in the Bay and suggested there may be mechanisms governing phytoplankton and DO over short time periods that current process models may not be able to capture. The upper reaches of the Bay are predominately affected by tributary influence. The well-mixed/weakly stratified nature of the Bay is evident in the strong influence of the sediment layer and deep waters on DO in the water column.

**Supplementary Materials:** The following supporting information can be downloaded at: https://www.mdpi.com/article/10.3390/w15061204/s1. S1. Model Input, S2. Observed Data, S3. Data Processing, S1.1. Tributary Inflow, S1.2. Tributary Boundary Conditions, S1.3. Benthic Flux, S1.4. Atmospheric Deposition, S1.5. Meteorology, Figure S1. Tributary Flow for Blackstone River, RI, US USGS 01113895, Table S1, Ammonia (as N), Table S2. Nitrate + Nitrite (as N), Table S3. Dissolved Organic Nitrogen (as N), Table S4. Dissolved Organic Nitrogen (as N), Table S5. Dissolved Organic P (as P), Table S6. Dissolved Oxygen, Table S7. Detrital C (as C), Table S8. Dissolved Oxygen, Table S9. Nitrogen Deposition, Table S10. Light. (Reference [47] is cited in the Supplementary Materials).

**Funding:** This work was funded through the US EPA's Office of Research and Development's Safe and Sustainable Water Research Program Product SSWR 5.2.2. This research did not receive any specific grant from funding agencies in the commercial or not-for-profit sectors.

**Data Availability Statement:** All observed data (depth profile and sonde data) and WASP simulated output (DO, chl *a*, and mechanistic scenarios) will be made available at: https://zenodo.org/record/7674989#.ZBMzpbTMKAQ (accessed on 24 February 2023). Data are currently embargoed until passed EPA clearance.

**Acknowledgments:** This work acknowledges the thoughtful and useful reviews of two anonymous reviewers, Autumn Oczkowski, Brandon Jarvis, Brenda Rashleigh, Betty Kreakie, and Marty Chintala, as well as conversations and discussions about the Narragansett Bay and previous modeling efforts with Hal Walker and Ed Dettmann.

**Conflicts of Interest:** The author declares no conflict of interest.

## Appendix A

**Table A1.** Constants and Parameters.

| Constant [a] | System Name | Value [b] | WASP Suggested Range |
|---|---|---|---|
| **Inorganic Nutrients** | | | |
| Nitrification Rate Constant | Ammonia | $0.1 \text{ d}^{-1}$ | 0–0.4 |
| Denitrification Rate Constant | Nitrate/Nitrite | $0.09 \text{ d}^{-1}$ | 0–0.4 |
| Nitrification Temperature Coefficient | Ammonia | 1.07 | 1.04–1.1 |
| Denitrification Temperature Coefficient | Nitrate/Nitrite | 1.04 | 1.04–1.1 |
| Minimum Temperature for Nitrification Reaction | Ammonia | $4 \,°C$ | 0–20 |
| Half Saturation Constant for Nitrification Oxygen Limit | Ammonia | $2 \text{ mg O}_2 \text{ l}^{-1}$ | 0–5 |
| Half Saturation Constant for Denitrification Oxygen Limit | Nitrate/Nitrite | $0.1 \text{ mg O}_2 \text{ l}^{-1}$ | 0–5 |
| Ammonia Partition Coefficient to Water Column Solids | TSS | $100 \text{ L kg}^{-1}$ | 0–200 |
| Orthophosphate Partition Coefficient to Water Column Solids | TSS | $100 \text{ L kg}^{-1}$ | 0–200 |
| **Organic Nutrients** | | | |
| Detritus Dissolution Rate | Detrital Carbon | $0.1 \text{ d}^{-1}$ | 0.01–0.2 |
| Dissolved Organic Nitrogen Mineralization Rate Constant | Dissolved Organic Nitrogen | $0.1 \text{ d}^{-1}$ | 0.01–0.2 |
| Dissolved Organic Phosphorous Mineralization Rate Constant | Dissolved Organic Phosphorous | $0.25 \text{ d}^{-1}$ | 0.01–0.22 |
| Temperature Correction for Detritus Dissolution | Detrital Carbon | 1.07 | 1.04–1.1 |
| Temperature Coefficient for Dissolved Organic Nitrogen Mineralization | Dissolved Organic Nitrogen | 1.07 | 1.04–1.1 |
| Dissolved Organic Phosphorous Mineralization Rate Constant | Dissolved Organic Phosphorous | 1.07 | 1.04–1.1 |
| **CBOD** | | | |
| CBOD Decay Rate Constant | CBOD-1 | $0.18 \text{ d}^{-1}$ | 0.05–0.4 |
| BOD Decay Rate Constant | CBOD-2 | $0.3 \text{ d}^{-1}$ | 0.05–0.4 |
| CBOD Decay Rate Temperature Correction Coefficient | CBOD-1 | 1.05 | 1–1.07 |
| BOD Decay Rate Temperature Correction Coefficient | CBOD-2 | 1.05 | 1–1.07 |
| CBOD Half Saturation Oxygen Limit | CBOD-1 | $0.5 \text{ mg O}_2 \text{ l}^{-1}$ | 0–0.5 |
| BOD Half Saturation Oxygen Limit | CBOD-2 | $0.5 \text{ mg O}_2 \text{ l}^{-1}$ | 0–0.5 |
| Fraction of Detritus Dissolution to CBOD | CBOD-1 | 0 | 0–1 |
| Fraction of Detritus Dissolution to BOD | CBOD-2 | 1 | 0–1 |
| Fraction of CBOD Carbon Source for Denitrification | CBOD-1 | 0.5 | 0–1 |
| Fraction of BOD Carbon Source for Denitrification | CBOD-2 | 0.5 | 0–1 |

**Table A1.** *Cont.*

| | Constant [a] | System Name | Value [b] | WASP Suggested Range |
|---|---|---|---|---|
| Phytoplankton | Phytoplankton Maximum Growth Rate Constant | Phytoplankton | 2.5 $d^{-1}$ | 0.5–4.0 |
| | Phytoplankton Respiration Rate Constant | Phytoplankton | 0.1 $d^{-1}$ | 0.05–0.25 |
| | Phytoplankton Death Rate Constant (Non-Zooplankton Predation) | Phytoplankton | 0 $d^{-1}$ | 0–0.1 |
| | Phytoplankton Growth Temperature Correction Coefficient | Phytoplankton | 1.07 | 1.05–1.08 |
| | Shape Parameter for Below Optimal Temperatures | Phytoplankton | 0.02 | 0.0–1.0 |
| | Shape Parameter for Above Optimal Temperatures | Phytoplankton | 0.02 | 0.0–1.0 |
| | Phytoplankton Optimal Light Saturation as PAR | Phytoplankton | 300 $W\,m^{-2}$ | 300–350 |
| | Phytoplankton Respiration Temperature Coefficient | Phytoplankton | 1.045 | 1.0–1.08 |
| | Phytoplankton Half Saturation Constant for Mineralization Rate | Phytoplankton | 0.8 mg phyt C $l^{-1}$ | 0–1 |
| | Phytoplankton Half Saturation Constant for N Uptake | Phytoplankton | 0.05 mg N $l^{-1}$ | 0.005–0.40 |
| | Phytoplankton Half Saturation Constant for P Uptake | Phytoplankton | 0.001 mg P $l^{-1}$ | 0.001–0.08 |
| | Fraction Phytoplankton Respiration Recycled to Organic N | Phytoplankton | 0.2 | 0–1 |
| | Fraction Phytoplankton Respiration Recycled to Organic P | Phytoplankton | 0.2 | 0–1 |
| | Fraction Phytoplankton Death Recycled to Detritus N | Phytoplankton | 0.8 | 0–1 |
| | Fraction Phytoplankton Death Recycled to Detritus P | Phytoplankton | 0.8 | 0–1 |
| | Phytoplankton Detritus to Carbon Ratio | Phytoplankton | 3.5 mg D:mg C | 2–5 |
| | Phytoplankton Nitrogen to Carbon Ratio | Phytoplankton | 0.16 mg N:mg C | 0.15–0.25 |
| | Phytoplankton Phosphorous to Carbon Ratio | Phytoplankton | 0.03 mg P:mg C | 0.01–0.05 |
| | Phytoplankton Carbon to Chlorophyll Ratio | Phytoplankton | 50 mg C:mg Chl | 25–125 |
| | Phytoplankton Settling Velocity | Phytoplankton | 0.01 $m\,d^{-1}$ | 0.005–0.2 |
| Sediments and Solids | Solid Settling Velocity | TSS | 0.2 $m\,d^{-1}$ | 0–2 |
| | Benthic Ammonia Flux | Ammonia | 7.78 mg N $m^{-2}\,d^{-1}$ | Observed [c] |
| | Sediment Oxygen Demand | Dissolved Oxygen | 1.12 mg $O_2$ $m^{-2}\,d^{-1}$ | Observed [c] |

Notes: [a] Rate constants are at 20 °C. [b] Constants and parameters based on ranges appropriate for the region, previous research [28,35,36], literature values [48], and WASP documentation, lecture notes, and user's guides (available at https://www.epa.gov/ceam/water-quality-analysis-simulation-program-wasp, accesses on 25 February 2023; http://epawasp.twool.com/, accesses on 25 February 2023). Italicized parameters are default values from WASP, and the non-italicized are specific to Narragansett Bay either through observation or calibration. WASP does not provide default values for Benthic Ammonia Flux and Sediment Oxygen Demand. [c] [47].

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
