# Peer review of "Simulating Hypoxia in a New England Estuary: WASP8 Advanced Eutrophication Module (Narragansett Bay, RI, USA)"

_water, doi:10.3390/w15061204_

Round 1
Reviewer 1 Report
Dear Author!
In your MS Simulating Hypoxia in a New England Estuary: WASP8 Ad-2 vanced Eutrophication Module (Narragansett Bay, USA) you deal with an interesting topic and question and is indeed worth publishing in Water, however, three major and multiple specific questions must be addressed.
Major comments:
1. Introduction:
Besides specific suggestions to improve the introduction (please see the attached pdf) I would suggest to extended the methodology part of the introduction, to better outline the niche you aim to fill with your work. This question will come up in the 'Discussion' as well.
2. Methods
· Site description: Please also add a paragraph on the pressures to the studied system.
· The model description is quite long, which is not a problem, however, it would definitely benefit the MS if a big chunk of it would go into supplement, mostly the really technical parts, and only those were kept, which are necessary to understand the results.
· In this rather long section (2.2-2.6) the information on gets lost
-why exactly were these parameters chosen? Were there others discarded?
-why were these sites chosen?
-are the chosen sites representative etc?
-It is not explained why only 2009 is explored. My question is, would you get a similar result for other years as well? Why is 2009 so special in the question of DO?
· Lines 351-355 should go into the Methods part. As I mentioned before, it should be more to the point in the main text, rather showing the steps and the rationale of the procedure than the specific details. The technical parts should go into the supplement as e.g. here
https://www.mdpi.com/2073-4433/13/1/93/htm this is just an example, do not cite the paper.
For example, the NS error only comes up here in the results. Nash JE, Sutcliffe JV (1970) River flow forecasting through conceptual models part I - a discussion of principles. J Hydrol 10(3):280–290. https://doi.org/10.1016/0022-1694(70)90255-6
In addition, I would suggest considering Lin's concordance correlation coefficient which gives greater insight into the prediction performance.
3. Results and discussion
On the 9 pages of the Results and discussion section there is not a single reference. Basically the discussion does not exist in the MS. The results should be discussed on an international scale. Are there similar settings, cases in the world, were researchers obtained similar results? How ere these handled, why is your approach unique etc. Are you findings in harmony with literature, I mean even process-wise? There is no clarification of the novum of the study because it is not discussed. These questions must be answered!
Please split the Results and Discussion into two sections.
Results sect. just present what you found,
Discussion: (a) Try to explain "What do your results mean?", and (b) how they relate to the literature,
Conclusions Re-state the main points in a new concise way that you want your readers to remember.
Overall, the study is worthy of publishing, but the international context has to be broadened, the methodology part has to be thoroughly restructured to rather show the flow and logic of the steps with the technical parts in supplement and the Results and Discussion section split and a proper discussion included.
The MS in tis present form is too long and it is not focused enough. By the time the reader finishes it she/he already lost the thread and the take home message. I would urge Authors to substantially decrease the length of the MS, primarily by placing text and figures into the supplement and try to focus on one thing? The take home message that resonates with the aims. If this is achieved, the MS will be ready for publishing.
I have added a commented pdf to the system. Please download it and extract my suggestions/questions and anwer those one-by-one.
Thank you!

Author Response
In your MS Simulating Hypoxia in a New England Estuary: WASP8 Advanced Eutrophication Module (Narragansett Bay, USA) you deal with an interesting topic and question and is indeed worth publishing in Water, however, three major and multiple specific questions must be addressed.
Response: Thank you for taking the time to provide a thoughtful review, with such great constructive criticism. I have gone through and addressed all of your comments below. I have attached the PDF with direct responses as well as have included a track changes document to show how this work has been incorporated into the document. These comments have been highly helpful in improving the quality and presentation of the paper as well as improving its impact. I particularly focused on improving the Introduction and Results and expanded the work to reflect the international relevance. I have included a PDF of the specific in-text comments with responses.
Major comments:
- Introduction:
Besides specific suggestions to improve the introduction (please see the attached pdf) I would suggest to extended the methodology part of the introduction, to better outline the niche you aim to fill with your work. This question will come up in the 'Discussion' as well.
Response: I agree with your assessment and have reworked the Introduction. I have added a statement to broaden the impacts of water quality to include all water bodies and brought up the issue of land use land cover change, particularly urbanization and agricultural development. The methodology was expanded to provide better detail on how this study was conducted and a final paragraph was added to provide a concise list of the what this study serves to achieve.
- Methods
- Site description: Please also add a paragraph on the pressures to the studied system.
Response: The primary stress on the system is via release of nutrients from WWTPs. Afew sentences were added to the first paragraph to highlight what this exactly entails:
Urban development surrounds the Bay, and Rhode Island is the second most densely populated (1,018 persons per sq. mi. in 2010) state in the US [29], primarily bordering the Upper Bay near Providence (Figure 1). There are 37 WWTPs in the watershed, with 11 WWTPs releasing directly into the Bay [17]. Approximately 2.05 x106 kg N yr-1 enters the Bay, with approximately 80% of that coming from WWTPs [30].
Agriculture has a minor impact, as there is only 5% agriculture in the watershed and its far away from the Bay itself. The main issue is that the watershed is highly urbanized in the upper reaches and there are 37 WWTPs in the watershed (with 11 directly releasing into the Bay).
- The model description is quite long, which is not a problem, however, it would definitely benefit the MS if a big chunk of it would go into supplement, mostly the really technical parts, and only those were kept, which are necessary to understand the results.
- In this rather long section (2.2-2.6) the information on gets lost
-why exactly were these parameters chosen? Were there others discarded?
-why were these sites chosen?
-are the chosen sites representative etc?
Response: The model input and observed data sections were all moved to the SI. A statement for why these 15 state variables were chosen is included:
The Advanced Eutrophication module, used here, specifically defines state variables (e.g., nitrate, ammonia, dissolved organic phosphorous). For the Narragansett Bay WASP model, 15 state variables were selected and simulated (see Table 1), capturing the full extent of WASP’s DO, nutrient cycling, and phytoplankton dynamics.
A statement for why these 5 sites were chosen was added:
Observations were taken at Phillipsdale, Bullock Reach, Conimicut point, North Providence, and Quonset Point (Figure 2). These locations were chosen because they capture the transect from the top of the Bay in the Seekonk River, downstream of a WWTP, along the Providence River, down the West Passage.
-It is not explained why only 2009 is explored. My question is, would you get a similar result for other years as well? Why is 2009 so special in the question of DO?
Response: 2009 was chosen as the study year because it was the year with the worst observed hypoxia. It was a wet year with high tributary flows. A single year was chosen to start. Future work will serve to expand this time frame. EFDC and WASP take a long time to run for so many segments with such short time steps and 15 state variables. Future work will serve to simulate longer years to address exactly the question you’re asking.
- Lines 351-355 should go into the Methods part. As I mentioned before, it should be more to the point in the main text, rather showing the steps and the rationale of the procedure than the specific details. The technical parts should go into the supplement as e.g. here
https://www.mdpi.com/2073-4433/13/1/93/htm this is just an example, do not cite the paper.
For example, the NS error only comes up here in the results. Nash JE, Sutcliffe JV (1970) River flow forecasting through conceptual models part I - a discussion of principles. J Hydrol 10(3):280–290. https://doi.org/10.1016/0022-1694(70)90255-6
In addition, I would suggest considering Lin's concordance correlation coefficient which gives greater insight into the prediction performance.
Response: I had never heard of Lin’s CCC. I would like to do more research and understand its strengths and limitations before incorporating it. It does sound interesting and useful and something I would like to include in future modelign studies. I have moved the mention of the metrics to the Methods, and the data processing required in Matlab is moved to the Supplemental Information.
- Results and discussion
On the 9 pages of the Results and discussion section there is not a single reference. Basically the discussion does not exist in the MS. The results should be discussed on an international scale. Are there similar settings, cases in the world, were researchers obtained similar results? How ere these handled, why is your approach unique etc. Are you findings in harmony with literature, I mean even process-wise? There is no clarification of the novum of the study because it is not discussed. These questions must be answered!
Please split the Results and Discussion into two sections.
Results sect. just present what you found,
Discussion: (a) Try to explain "What do your results mean?", and (b) how they relate to the literature,
Response: This is a great idea. I usually separate Results from Discussion in papers. I have created a Discussion section. In the new Discussion, I discuss the behavior of the Bay and how the model and observations support the formation of hypoxia. In this work, I reference modeling international studies, particularly the Pearl River, East China Sea, Baltic Sea, and Black Sea. I additionally reference the work in the Gulf of Mexico. I believe this strengthens the presentation and places the Narragansett Bay in the context of the wider field of estuary science.
The observations and WASP model simulations show the rise and fall of DO for the five sites of interest over the course of the year. The upper reaches (Phillipsdale and Bullock Reach) are typically stratified, with density driven circulation patterns, the middle section of the Bay (Conimicut Point and North Prudence) is weakly stratified or mixed, and the lower Bay (Quonset Point) is well-mixed [26]. The DO and phytoplankton observations and simulations reflect this structure (Figure 4, 5, and 8). Farthest upstream (Phillipsdale), hypoxia develops near the sediments early in the spring and then moves up the water column during summer until returning in autumn. Additionally, phytoplankton blooms (both observed and simulated) occurred in spring and then autumn. These results align with hypoxia driven by tributary nutrient loads [40]. Bullock Reach hypoxia begins later than at Phillspdale and penetrates up towards the surface. Conimicut Point, North Prudence, and Quonset develop hypoxia later, with shorter durations and less penetration towards the surface.
Fennel and Testa proposed an approach for relating hypoxia timescale to residence time as a 1:1 ratio [41]. The upper reach of the Bay (Providence-Seekonk River) has a residence time of 0.8 d to 13 d [42], which compares to the Pearl River, China (4 d) and the East China Sea (11 d) [41]. The upper reach behaves similarly to both of these systems, exhibiting summer hypoxia driven by sediment oxygen demand [43]. Using the approximate 1:1 relationships hypoxia timescale to residence time metric, the simulations support that upper reaches hypoxia faster than farther downstream segments. Using their designations as well, this study suggests that the upper reaches are more river-dominated systems compared to the middle and lower Bay. The main body of the Narragansett Bay has a residence time of 26 d (1.67 to 42.5 d) [26,42], which compares to the Northern Gulf of Mexico (30 d), while the Baltic Sea has a much higher residence time (3,100 d) [41]. In the Gulf of Mexico, the Mississippi River delivers large volume of freshwater carrying high nutrient loads, which results in a thin layer of hypoxia near the sediments [44]. The Baltic Sea exhibits permanent stratification with large zones of hypoxia [45]. Narragansett Bay is different from these system in that it does not exhibit strong or permanent stratification; the thickness of hypoxia is location and time dependent (Figure 8). The estimated small hypoxia timescale supports the quick rise and decline of hypoxia in the Bay, which may suggest management strategies to reduce nutrients released into the Bay may have a faster response for recovery. Recent research has suggested that improvements to WWTPs has resulted in decreased hypoxia, nutrient concentrations, and phytoplankton growth [19,32].
The effect of wind can disrupt hypoxia, as seen in the Gulf of Mexico where high wind events mix the water column. Hypoxia in the Gulf is reestablished quickly when the wind-induced mixing subsides [44]. Recent modeling has also shown wind-driven hypoxia, where winds caused bottom water upwelling, pushing hypoxic waters to the nearshore [22]. The Bay is relatively protected from winds, and while strong wind events can impact flows in the passages, generally wind driven effects are minimal and serve mainly to promote mixing [26]. Some observations have suggested possible upwelling in Bullock Reach in the Providence River (personal communication, Narragansett Bay Estuary Program).
In this model, CBOD was simulated as two different parameters. Recent research has suggested that terrestrial CBOD may be subject to photochemical reactions, which can provide nutrients for microbial communities [46]. Currently WASP does not incorporate this process, which could potentially account for increased phytoplankton growth. This could in turn account for the large diurnal variations in DO and phytoplankton concentrations, which the WASP model is not currently able to simulate.
The penetration of hypoxic waters from the deep waters into the water column suggests the importance of oxygen demand and nutrient fluxes from the sediment (Figure 8 and 9). The well-mixed and weakly stratified structure of the Bay supports this results. In the
Baltic Sea, the role of sediments were found to change with DO levels in overlying waters, shifting from nitrogen removal to nitrogen release as hypoxia worsens [45]. As nutrient reduction management strategies are put into place, it is unclear what the response of the sediments will be and how fast changes may occur. The presence of legacy nutrients in the Bay may result in a lag in the water quality response and some zones of the Bay may change at different rates than other zones. Looking to long term recovery of the Bay, it will be important to incorporate sediment diagenesis processes with adequate parameterization to investigate the effect of management strategies in the Bay watershed as well as implications for land-use and climate change.
Conclusions Re-state the main points in a new concise way that you want your readers to remember.
Response: A new concise Conclusions has been included focuses solely on the take home messages.
A 3-dimensional mechanistic fate and transport model was developed for the year 2009 for the Narragansett Bay. The model helps inform our understanding on hypoxia and serves as a foundation for further studies, identifying scientific research needs as well as the potential for investigating longer time frames, climate change, LULC change, and acidification. The model was able to capture the trends and patterns of DO and phytoplankton in the Bay and suggested there may be mechanisms governing phytoplankton and DO over short time periods that current process models may not be able to capture. The upper reaches of the Bay are predominately affected by tributary influence. The well-mixed/weakly stratified nature of the Bay is evident in the strong influence of the sediment layer and deep waters on DO in the water column.
Overall, the study is worthy of publishing, but the international context has to be broadened, the methodology part has to be thoroughly restructured to rather show the flow and logic of the steps with the technical parts in supplement and the Results and Discussion section split and a proper discussion included.
Response: Thank you for comments. I appreciate the constructive criticism that serves to strengthen this paper.
The MS in tis present form is too long and it is not focused enough. By the time the reader finishes it she/he already lost the thread and the take home message. I would urge Authors to substantially decrease the length of the MS, primarily by placing text and figures into the supplement and try to focus on one thing? The take home message that resonates with the aims. If this is achieved, the MS will be ready for publishing.
I have added a commented pdf to the system. Please download it and extract my suggestions/questions and anwer those one-by-one.
Thank you!

Reviewer 2 Report
The present manuscript applies a mechanistic model to understand hypoxia phenomenon in the Narragansett Bay (USA). The implementation of the model is based on a very large set of environmental parameters, the methods are described properly and the manuscript is well written.
However, the mechanisms of coastal hypoxia are not investigated in depth and the author does not seem to compare the outputs of this model with the scientific literature available for this phenomenon. For this reason, I think that the Discussion of this manuscript should be improved with a MINOR revision addressed to the following scientific points:
Mechanisms of coastal hypoxia formation:
Hypoxia in the coastal systems can originate by complex combinations of environmental forcings. For example:
· Periods of low runoff, characterized by stable meteorological conditions, weak renewal of the coastal waters, persistent stratification of the water column and a high benthic respiration that causes DO depletion in the deeper waters (summer hypoxia);
· Events of high runoff, characterized by large phytoplankton blooms triggered by river nutrient loads, in particular if the seawater temperature is still high and the sinking of the biomass is fast, due to a weak stratification of the water column (coastal hypoxia, mainly spring and autumn);
· Landward winds pushing eutrophic river plumes against the coast for long periods (wind-driven hypoxia) or seaward wing causing the upwelling of offshore bottom hypoxic waters along the coast (upwelling-driven hypoxia).
The author should discuss if the model results suggest the occurrence of any of these mechanisms of hypoxia formation in the Narragansett Bay.
Effects of sediment inputs on DO concentration (Lines 459-464):
The authors should consider here that primary production in the coastal waters is often limited by a scarce penetration of the light in the water column, due to the increase of the turbidity generated by the advection of river sediments. The increase of DO concentrations in the case “No Sediment Inputs” might originate by an increased photosynthesis in the water column, rather than by changes in the sediment nutrient fluxes.
Photo-chemical degradation of terrestrial organic matter (Line 40-45):
DO concentration can vary not only because of the discharge of inorganic nutrients, but also because of the discharge of terrestrial organic matter. Terrestrial organic pool is usually considered to be refractory, but it can be degraded by photo-chemical reactions induced by the sunlight. This process can produce both nutrients for primary producers or semi-labile organic matter enhancing the respiration of microbial community (e.g. Schafer et al. 2021, Biogeochemistry 152,291–307, Doi: 10.1007/s10533-021-00756-0). Might this process concur to amplify the diurnal oscillations of DO concentration in the observed time series compared to the model results (Figure 4) ?
Description of the environmental conditions during the reference year 2009:
The model used in this study includes a large set of hydrological, biogeochemical and meteorological parameters that affect DO balance in the Narragansett Bay. However, the conditions in the bay during the reference year 2009 might be better summarized for the readers showing the annual cycle of some of the most important components of this coastal system: total freshwater load in the bay, residence time of the water in the bay (if available), total nitrogen and phosphorus loads in the bay, stratification of the water column (at selected locations), total plankton biomass in the bay (if available). This information might be provided in the Supplementary Materials using figures (plots with the time series of these parameters in 2009) and/or tables and briefly mentioned in the text.

Author Response
The present manuscript applies a mechanistic model to understand hypoxia phenomenon in the Narragansett Bay (USA). The implementation of the model is based on a very large set of environmental parameters, the methods are described properly and the manuscript is well written.
However, the mechanisms of coastal hypoxia are not investigated in depth and the author does not seem to compare the outputs of this model with the scientific literature available for this phenomenon. For this reason, I think that the Discussion of this manuscript should be improved with a MINOR revision addressed to the following scientific points:
General Response: Thank you for your thoughtful and insightful comments. They have been of great use. I have considered all of them and incorporated details on your comments within the paper as seen below. The incorporation of these comments have improved the presentation and overall impact of the paper.
Mechanisms of coastal hypoxia formation:
Hypoxia in the coastal systems can originate by complex combinations of environmental forcings. For example:
- Periods of low runoff, characterized by stable meteorological conditions, weak renewal of the coastal waters, persistent stratification of the water column and a high benthic respiration that causes DO depletion in the deeper waters (summer hypoxia);
- Events of high runoff, characterized by large phytoplankton blooms triggered by river nutrient loads, in particular if the seawater temperature is still high and the sinking of the biomass is fast, due to a weak stratification of the water column (coastal hypoxia, mainly spring and autumn);
- Landward winds pushing eutrophic river plumes against the coast for long periods (wind-driven hypoxia) or seaward wing causing the upwelling of offshore bottom hypoxic waters along the coast (upwelling-driven hypoxia).
The author should discuss if the model results suggest the occurrence of any of these mechanisms of hypoxia formation in the Narragansett Bay.
Response: These are very good points and quite helpful in framing the work that has been done with this effort, particularly with framing it within the context of the complex processes and combinations of process that can cause hypoxia. The Discussion section has been reworked. Within the Discussion, these topics have been additionally addressed. I particularly liked your idea of incorporating the different forcing functions. As there isn’t space to discuss everything, I have primarily focused on residence time and used that to reflect on Fennel and Testa’s (2019) work on relating a hypoxia timescale to residence time. I additionally address the issue on how the upper reaches are stratified with density circulation, the middle Bay is weakly stratified, and the lower Bay is well-mixed.
The observations and WASP model simulations show the rise and fall of DO for the five sites of interest over the course of the year. The upper reaches (Phillipsdale and Bullock Reach) are typically stratified, with density driven circulation patterns, the middle section of the Bay (Conimicut Point and North Prudence) is weakly stratified or mixed, and the lower Bay (Quonset Point) is well-mixed [26]. The DO and phytoplankton observations and simulations reflect this structure (Figure 4, 5, and 8). Farthest upstream (Phillipsdale), hypoxia develops near the sediments early in the spring and then moves up the water column during summer until returning in autumn. Additionally, phytoplankton blooms (both observed and simulated) occurred in spring and then autumn. These results align with hypoxia driven by tributary nutrient loads [40]. Bullock Reach hypoxia begins later than at Phillspdale and penetrates up towards the surface. Conimicut Point, North Prudence, and Quonset develop hypoxia later, with shorter durations and less penetration towards the surface.
Fennel and Testa proposed an approach for relating hypoxia timescale to residence time as a 1:1 ratio [41]. The upper reach of the Bay (Providence-Seekonk River) has a residence time of 0.8 d to 13 d [42], which compares to the Pearl River, China (4 d) and the East China Sea (11 d) [41]. The upper reach behaves similarly to both of these systems, exhibiting summer hypoxia driven by sediment oxygen demand [43]. Using the approximate 1:1 relationships hypoxia timescale to residence time metric, the simulations support that upper reaches hypoxia faster than farther downstream segments. Using their designations as well, this study suggests that the upper reaches are more river-dominated systems compared to the middle and lower Bay. The main body of the Narragansett Bay has a residence time of 26 d (1.67 to 42.5 d) [26,42], which compares to the Northern Gulf of Mexico (30 d), while the Baltic Sea has a much higher residence time (3,100 d) [41]. In the Gulf of Mexico, the Mississippi River delivers large volume of freshwater carrying high nutrient loads, which results in a thin layer of hypoxia near the sediments [44]. The Baltic Sea exhibits permanent stratification with large zones of hypoxia [45]. Narragansett Bay is different from these system in that it does not exhibit strong or permananet stratification;the thickness of hypoxia is location and time dependent (Figure 8). The estimated small hypoxia timescale supports the quick rise and decline of hypoxia in the Bay, which may suggest management strategies to reduce nutrients released into the Bay may have a faster response for recovery. Recent research has suggested that improvements to WWTPs has resulted in decreased hypoxia, nutrient concentrations, and phytoplankton growth [19,32].
The effect of wind can disrupt hypoxia, as seen in the Gulf of Mexico where high wind events mix the water column. Hypoxia in the Gulf is reestablished quickly when the wind-induced mixing subsides [44]. Recent modeling has also shown wind-driven hypoxia, where winds caused bottom water upwelling, pushing hypoxic waters to the nearshore [22]. The Bay is relatively protected from winds, and while strong wind events can impact flows in the passages, generally wind driven effects are minimal and serve mainly to promote mixing [26]. Some observations have suggested possible upwelling in Bullock Reach in the Providence River (personal communication, Narragansett Bay Estuary Program).
Effects of sediment inputs on DO concentration (Lines 459-464):
Comment: The authors should consider here that primary production in the coastal waters is often limited by a scarce penetration of the light in the water column, due to the increase of the turbidity generated by the advection of river sediments. The increase of DO concentrations in the case “No Sediment Inputs” might originate by an increased photosynthesis in the water column, rather than by changes in the sediment nutrient fluxes.
Response: This is a good point. The following statement is added to Section 3.4: Without sediment nutrient fluxes, water clarity could improve allowing for increased light penetration in the water column, potentially increasing photosynthesis and DO.
Photo-chemical degradation of terrestrial organic matter (Line 40-45):
DO concentration can vary not only because of the discharge of inorganic nutrients, but also because of the discharge of terrestrial organic matter. Terrestrial organic pool is usually considered to be refractory, but it can be degraded by photo-chemical reactions induced by the sunlight. This process can produce both nutrients for primary producers or semi-labile organic matter enhancing the respiration of microbial community (e.g. Schafer et al. 2021, Biogeochemistry 152,291–307, Doi: 10.1007/s10533-021-00756-0). Might this process concur to amplify the diurnal oscillations of DO concentration in the observed time series compared to the model results (Figure 4) ?
Response: This is an interesting point I had not considered. The Narragansett Bay model had two types of BOD, one which degrades slower and one that degrades faster. Currently I didn’t set up the model to have photo-chemical reactions which degrade organic matter. This is something I will need to investigate if WASP has the capability to manage that. If not, I could work on incorporating this process into the model itself for future studies. I have added a citation to Schafer et al., 2021 and mentioned that this could improve model simulations for the diurnal DO oscillations.
In this model, CBOD was simulated as two different parameters. Recent research has suggested that terrestrial CBOD may be subject to photochemical reactions, which can provide nutrients for microbial communities [46]. Currently WASP does not incorporate this process, which could potentially account for increased phytoplankton growth. This could in turn account for the large diurnal variations in DO and phytoplankton concentrations, which the WASP model is not currently able to simulate.
Description of the environmental conditions during the reference year 2009:
Comment: The model used in this study includes a large set of hydrological, biogeochemical and meteorological parameters that affect DO balance in the Narragansett Bay. However, the conditions in the bay during the reference year 2009 might be better summarized for the readers showing the annual cycle of some of the most important components of this coastal system: total freshwater load in the bay, residence time of the water in the bay (if available), total nitrogen and phosphorus loads in the bay, stratification of the water column (at selected locations), total plankton biomass in the bay (if available). This information might be provided in the Supplementary Materials using figures (plots with the time series of these parameters in 2009) and/or tables and briefly mentioned in the text.
Response: These are really useful and interesting ideas. I’ve incorporated information on phytoplankton, Residence time,
Phytoplankton: Total phytoplankton biomass isn’t available, but there is information for specific locations. This was added for context: “At the Providence River, mean [chl a] was 22 ug l-1 and in the Lower Bay, mean [chl a] was 5 ug l-1” was added to provide details on phytoplankton in the Bay.
Residence time and Freshwater flow information was added: mean residence time of 26 d (range 1.67 d to 42.5 d), and annual freshwater input of 5 x 106 m3 yr-1 [26-28].
Residence time was a focus in the new Discussion was added on residence time, with additional separation of information on residence time in the upper reaches of the Bay (Providence-Seekonk River) is 0.8 to 13 d.
Nitrogen loading:
Urban development surrounds the Bay, and Rhode Island is the second most densely populated (1,018 persons per sq. mi. in 2010) state in the US [29], primarily bordering the Upper Bay near Providence (Figure 1). There are 37 WWTPs in the watershed, with 11 WWTPs releasing directly into the Bay [17]. Approximately 2.05 x106 kg N yr-1 enters the Bay, with approximately 80% of that coming from WWTPs [30].
There isn’t solid information on phosphorous loading to the Bay that I could find that is rigorous enough to include.
Round 2
Reviewer 1 Report
After the revisions, the paper is now acceptabel for publication.